# Spatially distributed impacts of climate change and groundwater demand on the water resources in a Wadi system

Nariman Mahmoodi[1], Jens Kiesel[1,2], Paul D. Wagner[1], Nicola Fohrer[1]

[1]Kiel University, Department of Hydrology and Water Resources Management, Kiel 24118, Germany

[2]Leibniz-Institute of Freshwater Ecology and Inland Fisheries, Department of Ecosystem Research, Berlin 12489, Germany

*Correspondence to*: Nariman Mahmoodi (nmahmoodi@hydrology.uni-kiel.de)

**Abstract.** Understanding current and possible future alterations of water resources under climate change and increased water demand allows for better water and environmental management decisions in arid regions. This study aims at analyzing the impact of groundwater demand and climate change on groundwater sustainability and hydrologic regime alterations in a Wadi system in central Iran. A hydrologic model is used to assess streamflow and groundwater recharge of the Halilrood Basin on a daily time step under five different scenarios over baseline period (1979-2009) and for two future scenario periods (near future: 2030–2059 and far future: 2070-2099). The Indicators of Hydrologic Alteration (IHA) with a set of 32 parameters are used in conjunction with the Range of Variability Approach (RVA) to evaluate hydrologic regime change in the river. The results show that groundwater recharge is expected to decrease, and is not able to fulfil the increasing water demand in the far future scenario. The Halilrood River will undergo low and moderate streamflow alteration under both stressors during the near future as RVA alteration is classified as "high" for only three indicators, whereas stronger alteration is expected in the far future with 11 indicators in the "high" range. Absolute changes in hydrologic indicators are stronger when both climate change and groundwater demands are considered in the far future simulations, since 27 indicators show significant changes and RVA show high and moderate levels of changes for 18 indicators. Considering the evaluated RVA changes, future impacts on the freshwater ecosystems in the Halilrood Basin will be severe. The developed approach can be transferred to other Wadi regions for a spatially-distributed assessment of water resources sustainability.

## 1 Introduction

Water resources are important in arid regions and any alteration caused by anthropogenic activities might have strong environmental and socio-economic impacts. This poses a serious threat to the sustainable development of water resources in different sectors (Oki and Kanae, 2006 and Panahi et al., 2020). Hence, sustainable management of water resources is vital especially in arid regions with limited water availability (Wu et al., 2013; Davijani et al., 2016; Yu et al., 2019).

Sustainable use of water resources should be jointly assessed with regard to surface water and groundwater. Groundwater is not only a valuable source of high-quality freshwater and plays a central role in sustaining water supplies and rural livelihoods in arid regions (Giordano, 2009; Cuthbert et al., 2019), but also contributes to base flow and the functioning of freshwater

ecosystems (Boulton and Hancock, 2006; Kath et al., 2018). Excessive groundwater withdrawal for a wide variety of activities, is causing aquifers to rapidly deplete worldwide (Gleeson and Wada, 2013). Groundwater withdrawal has more severe consequences in arid and semi-arid regions, where surface water is insufficient to meet human water demand especially in times of droughts and natural groundwater recharge is low (Long et al., 2016; Taylor, 2014). Moreover, the existence of different and effective groundwater withdrawal systems such as qanats and wells in arid regions can lead to pronounced

groundwater depletion (Eissa et al., 2016; Perrone and Jasechko, 2019). Substantial and persistent drops in groundwater levels are expected when the ratio of groundwater demand exceed recharge from infiltration and river transmission losses over the basin (de Graff et al., 2019; Acero Triana et al., 2020). Therefore, the ratio of groundwater demand to the recharge rate is a potential indicator of regional water security (Richey et al., 2015) and sustainability. Little and sporadic precipitation, very high evaporation, little percolation and groundwater recharge are peculiar characteristics of Wadi regions (Pahlevani

Majdabady et al., 2020; Messerschmid et al., 2020). In Iran, groundwater extraction rates increased over the last decades due to the scarcity of precipitation, combined with climate change and population growth (Izady et al., 2015; Rafiei Emam et al., 2015; Mahmoudpour et al., 2016). While climate change impacts on groundwater resources are well understood, the combined effects of climate change and population growth (water demand) on groundwater resources are rarely analyzed in a spatially distributed manner. Therefore, estimating the current and future amount of average annual groundwater recharge and storage

under climate change conditions and by incorporating growing water demands due to development and population growth is critical and fundamental for a sustainable management of groundwater and surface water (Dash et al., 2019).

Moreover, hydrological changes caused by climate change and population growth are not limited to groundwater, but also extend to surface water resources, where changes in runoff timing, seasonality, peak rates and volumes of surface water have been reported for different arid parts of Iran (Ashraf et al., 2019) and other countries e.g., in the United States (Caldwell et al.,

2012), Algeria (Achite and Ouillon 2016), China (Xue et al., 2017), and Jordan (Al Qatarneh et al., 2018). Alterations of the streamflow regime can result in negative environmental consequences, as e.g., in China, where decreases in water resources had a negative effect on the semi-arid wetland ecosystem of Western Jilin (Moiwo et al., 2010). Wen et al., (2013) reported that reduction in streamflow is the principal cause of the decrease in ecological values of a semi-arid wetland in Australia. Similarly in the northwest of Iran, a dramatic reduction of the water level of Urmia lake has been reported by Khazaei et al.,

(2019) due to the reduced inflow to the lake from the entire basin. Moreover, the Bakhtegan and Tashk lakes in southern Iran started to disappear due to hydrologic regime changes in Kore River, which altered the inflow to the lakes (Haghighi and Kløve 2017). The fluctuation of streamflow in Hirmand basin caused several hydrologic and environmental effects such as a decrease in water level of Hamoun wetland, increasing wildlife death rates, and increasing air pollution and consequently health problems, in southwestern Iran (Sharifikia, 2012). In addition, Nielsen and Brock (2009) found a shift in species distribution

in wetlands of southern Australia due to streamflow regime alteration and salinity induced by climatic changes. According to Qaderi Nasab and Rahnema (2020), the Jazmorian wetland, which is fed by Wadis in central Iran, has undergone significant changes in area and seasonal availability of water between 1987 and 2017. In addition, they report very low soil moisture in the wetland area due to decreasing inflows and high potential evaporation (more than 2800 mm $yr^{-1}$), which increases

vulnerability of the wetland to wind erosion. Modarres and Sadeghi (2018) showed that the dust from the wetland increased
the number of dusty days in Iranshahr city, which is almost 180 km away from the wetland. Vulnerability of wetlands to wind
erosion has also been found in other arid regions e.g., the dried-up Ebinur Lake region in northwestern China has become one
of the main dust sources as a consequence of the change of inflow to the lake (Bao et al., 2006). Further aggravation of climate
change will put increasing pressure on the already threatened natural ecosystem of Wadi regions. Therefore, future
susceptibility of Wadis to climate change and growing groundwater demand is important to understand.

Recognizing the above concerns, this study aims to: (1) assess the sustainability of groundwater in the future by modeling the
recharge rate under climate change and predicted withdrawals, (2) explore possible future hydrologic alterations of rivers in
Wadi regions and evaluate their ecological implications.

## 2 Materials and methods

### 2.1 Study area

The Halilrood Basin (7224 km$^2$) is located in central Iran (Figure 1a). It is a major river in the Kerman province in terms of
discharge, and provides various ecosystem services, as the water is used for domestic, industrial, energy (Jiroft Dam, Figure
1b), and agricultural (small scale farming) purposes, and it provides water to the Jazmorian wetland (Figure 1b), mainly from
February to April (Figure 1c). Annual average precipitation in Halilrood Basin varies between 121 mm to 511 mm with an
average of 295 mm from 1979 to 2011 (IWPCO, 2015). The annual potential evaporation is more than 2500 mm and the mean
annual discharge (7.68 m$^3$ s$^{-1}$) is about one tenth of the precipitation (IWPCO, 2015). Within the period 1979-2011 streamflow
intermittency at the outlet of the basin has increased most significantly in 2005 and 2007. Regarding land cover, bare land
areas occupy about 75% of the basin. According to Mahmoodi et al. (2020a), shrubland and grassland areas can be found in
the highlands, whereas irrigated agriculture is existing only in proximity to the rivers and water use systems (WUSs: qanats,
wells, springs). Three cities, i.e. Baft, Bazanjan, Rabor are located in the northern part of the basin (Figure 1d). Water from
shallow aquifer extracted through springs, qanats, and wells drilled in different parts of the basin (Figure 1e), is used to supply
water to the cities and villages mainly for drinking and washing and to small-scale farming for irrigation.

### 2.2 Hydrological model

The Soil and Water Assessment Tool (SWAT, Arnold et al., 1998; Arnold et al., 2012) is used to simulate the streamflow of
Halilrood River between 1993 and 2009 on a daily time step. SWAT is a semi-distributed model which is most commonly
applied to simulate water fluxes on the catchment scale with diverse agricultural management options and under various
hydrologic conditions over long periods of time (Arnold et al., 2012). As a process-based hydrological model SWAT has
proven its capability for climate change impact studies (Emami and Koch, 2019; Tigabu et al., 2020). The SWAT model of
the Halilrood Basin is divided into 285 sub-basins and 6091 hydrologic response units (HRUs) defined by land use (FAO,
2009), slope, and soil type (ESA, 2010). Based on an equal distribution of dry years (total precipitation < 200 mm), wet years

(total precipitation > 270 mm) and average years (200–270 mm annual precipitation) in the study area, an eight-year period of observed data provided by Iran Water & Power Resources Development Company (IWPCO, 2015) is used for model calibration (1995-2003) and a six-year period for validation (2004–2009). Both calibration and validation periods are composed of almost 1/3 dry, wet, and normal years, respectively. Water use systems (WUSs) and soil and water conservation measures (SWCMs) scattered within the basin were implemented in the model (Mahmoodi et al., 2020a). According to the

model performance rating suggested by Moriasi et al. (2007), very good and satisfactory performances for modeling daily streamflow were achieved judged by a multi-metric approach including NSE (0.76 and 0.54), PBIAS (4.7 and 7.1), RSR (0.49 and 0.78), and the modified KGE (0.87 and 0.62) for calibration and validation period, respectively. The calibrated hydrologic model showed also a good performance (NSE = 0.65) for simulating potential evaporation (PE) at the sub-basin scale, where the comparison showed a good agreement between simulated and observed PE at the synoptic station shown in Figure 1d. In

addition, modelled annual actual evaporation (AE) for the Halilrood Basin between 1995 and 2009 (min.:100.2 mm yr$^{-1}$, median: 173.1, max.: 274.2,) is in a similar range as the AE from the Global Land Evaporation Amsterdam Model (GLEAM version 3.5a, https://www.gleam.eu/; Martens et al., 2017; Miralles et al., 2011) (min.: 96.7 mm yr$^{-1}$, median: 163.1, max.: 255.9). Further, the groundwater recharge of the Halilrood basin estimated by SWAT is around 50 mm yr$^{-1}$, which is in agreement with the recharge rate reported by Parizi et al. (2020) for most Wadis in central Iran. No observations or estimates

of river bed infiltration are available for the Halilrood basin and it generally is one of the most challenging water balance components to be quantified in Wadi regions (Wheater et al., 2008; Neitsch et al., 2011). Given the plausible representation of all other water balance components in the model, it can be inferred that simulated bed infiltration is represented realistically. A more detailed model description and evaluation is available in Mahmoodi et al., (2020a).

### 2.3 Future climate change simulation

Mahmoodi et al., (2020b) used an ensemble of 17 global and regional climate models (G-RCMs) from the Coordinated Regional Climate Downscaling Experiment−CORDEX (Jacob et al., 2014) to assess the impact of future climate change on streamflow and major hydrological components of the Halilrood Basin. Climate data of the Representative Concentration Pathway (RCP) 8.5 were bias corrected with two methods (distribution mapping and linear scaling) and evaluated alongside the raw (not bias corrected) data. RCP8.5 is selected since actual green-house gas emissions of the last decade have followed

the RCP8.5 trajectory closer than any of the other RCPs (Sanford et al., 2014). Although the projections driven from scenarios with high CO2 emissions (e.g. RCP8.5) have been criticized as mitigation measures are not accounted for (Hausfather and Peters, 2020), the projections from these scenarios can still be valuable (IPCC, 2021) as they not only agree with historical total cumulative CO2 emissions, but are also plausible for future projections given current and stated policies (Schwalm et al., 2020). The climate model (G-RCM CSIRO-SMHI, RCP8.5, bias adjusted with distribution mapping; Figure 2a) is selected

according to the 'model democracy' or 'ensemble of opportunity' approach (IPCC 2013) that represents the median for most of the simulated hydrological components, i.e., evaporation, water yield, surface runoff, lateral flow, and groundwater flow (Mahmoodi et al., 2020b). This procedure of analyzing the impacts of all climate models in an ensemble on the target indicator

(here: streamflow) and then selecting the median model is one of many possible approaches in climate change impact studies (Kiesel et al., 2021). The climate models leading to min (CSIRO+IITM) and max (CCCma+SMHI) hydrological components are analysed in addition to the median model to quantify the uncertainty range associated with the full climate change ensemble (Figure 2a). Similar to the median model, this min and max analysis is carried out for all water use system scenarios (Table 5) The climatic conditions of the selected median climate model are within the range of conditions of the baseline period as the driest and wettest future years are already included in the baseline years. Therefore, it can be assumed that the parameterized SWAT model is sufficiently applicable the future climate conditions. The calibrated and validated SWAT model is run with the selected climate model output to simulate groundwater recharge and streamflow for the baseline period (1979-2009) and two future periods (near future: 2030-2059 and far future: 2070-2099). The choice of the baseline period can alter the depiction of the changes in hydrologic indicators under climate change, but its uncertainty is lower when baseline periods exceed multiple decades (Ruokolainen and Räisänen 2007). Basic statistical analysis of streamflow at the basin outlet for baseline period and future climate conditions (median, min, and max model) are shown in Table 1. The contradiction shown for min annual streamflow simulated for max (the wettest condition) and the median climate models might be due to the distribution of the rain day and the way of min and max climate model selection, which is based on simulated water balanced components.

## 2.4 Future population growth and water demand

Population growth is the main factor governing water consumption in Iran, as Keshavarz et al. (2006) reported a significant correlation between water consumption and population/size of households in Fars province. In addition, the water consumption data reported for three provinces i.e., Azarbaijan, Khuzestan, Isfahan during the period 2001-2010 shows that the consumption rate is increasing linearly with population growth (Mombeni et al., 2013). Based on the data reported by the Statistical Center of Iran (SCI, 2017), Iran has experienced a remarkable population increase within the last few decades (from 33.7 million (M) in 1976 to more than 80 M in 2017 (Dienel et al., 2017)). According to the last census in 2017, the total urban and rural population of Halilrood Basin was 124,000 (Statistical Center of Iran-SCI, 2017). Four population growth rate scenarios are suggested by PBO (2019), i: childbirth rate decreases with a steep slope, ii: childbirth rate decreases with a gentle slope, iii: childbirth rate remains constant, iv: childbirth rate increases. Among these scenarios, a conservative scenario, constant childbirth rate (scenario iii) —i.e. the current trend of population growth will remain constant in the future— is applied on the 2017 population data to estimate the population of the basin for the years 2045 and 2085, representative for the near and far future periods respectively (Table 2).

Future water demand in Halilrood Basin is projected by considering (i) groundwater withdrawal from WUSs and (ii) minimum and maximum water consumption for the estimated population.

(i): To meet the future domestic, agricultural and industrial water demand, increases in the number of wells and qanats are linearly extrapolated with the estimated increases in the population of Halilrood Basin as follows:

$$NWUSj = \frac{Pj}{Pi} * NWUSi, \tag{1}$$

Where NWUS$j$ and NWUS$i$ are the number of water use systems in the year $j$ and $i$, respectively; P$j$ and P$i$ is population in the year $j$ and $i$, respectively. The number of springs as a natural WUS is assumed to remain constant in the future. The annual average water withdrawal per WUS recorded for the baseline period is assumed to remain constant in the future and is used to linearly extrapolate the groundwater demand for each sub-basin for the future number of WUS (NWUS$j$) for 2045 and 2085 (Table 3). The number of WUSs are reported until 2011. Therefore, the population growth rate between 2011 and 2017 is used to determine the number of WUS in 2017 (Table 3).

(ii): The minimum and maximum amount of water required per person per day in Iran is about 135 and 300 litres, respectively (ISC, 2017-2018). According to these numbers and the estimated population growth (Table 2), maximum and minimum water consumption in near and far future are estimated (Table 4).

**2.5 Scenarios**

To disentangle the impacts of climate change and population growth and its combined effects on future aquifer condition and hydrologic regime, five scenarios are developed (Table 5, Figure 2b). "NO-WUS" scenario is included, to assess the sole impact of climate change on the hydrologic regime under pristine conditions. It therefore represents a scenario where all anthropogenic extractions have ceased. "Constant-WUS" scenario is defined to investigate the impact of climate change on hydrologic regime and groundwater sustainability in the future simulations in comparison to the current condition by keeping the number of WUSs unaltered. The impacts of both climate change and WUSs on groundwater sustainability and hydrologic regime are assessed under "Projected-WUS" scenario. To precisely indicate the impact of the sole water demand by the population on groundwater sustainability in near and far future, the maximum and minimum amount of water required per person is computed and considered under min- and max-consumption scenarios. These two scenarios are considered only on the entire basin scale due to limited availability of information regarding population growth on smaller scales (e.g., villages). Minimum and maximum water consumption is included in the Constant- and Projected-WUS scenarios.

**2.6 Groundwater sustainability**

Groundwater sustainability is assessed on two different spatial scales: on the sub-basin and on the entire basin scale.

**2.6.1 Sub-basin scale**

Groundwater sustainability (GWS) on the sub-basin scale is defined as the ratio of groundwater demand (GWD) to groundwater recharge (GWR) (Figure 2c).

To provide an appropriate estimate and range of the future aquifer condition on the sub-basin scale, groundwater demand for the baseline period and two future periods is estimated for two scenarios: Projected-WUS and Constant-WUS. Moreover, groundwater recharge is averaged for the entire 30-year periods.

## 2.6.2 Entire basin scale

The possible connection of groundwater bodies across sub-basins is considered by treating the Halilrood Basin as one integrated groundwater system. Therefore, groundwater sustainability (GWS) is assessed by comparing the total groundwater recharge (GWR) over the entire basin and entire 30-year periods to (i) the total projected groundwater demand (GWD) from the WUSs under Projected-WUS scenario, (ii) the minimum, and (iii) the maximum water consumptions (min- and max-WC) estimated for the growing population under min- and max-consumption scenarios.

## 2.7 Indicators of Hydrologic Alteration (IHA)

Changes in the hydrologic regime of the Halilrood River that are caused by climate change and growing groundwater demand are not only a challenge for the water sector (e.g., small-scale farming), but also decrease groundwater levels and threaten the Jazmorian wetland ecosystem by reducing its water availability. The hydrologic regime alteration is analyzed for the flow into the wetland under the following scenarios: No-WUS, Constant-WUS, and Projected-WUS (Figure 2c).

Numerous hydrologic indicators have been developed to describe different components of the streamflow regime. A set of 32 hydrologic indicators are used to assess changes in the hydrologic streamflow regime (Richter et al., 1996). The indicators are categorized into five groups; Group1: Magnitude of monthly water conditions, Group2: Magnitude of annual extreme streamflow events with different durations, Group3: Timing of annual extreme water conditions, Group4: Frequency and duration of high and low streamflow pulses, and Group5: Rate and frequency of water condition changes (Table 6). The "IHA" software developed by The Nature Conservancy (TNC, 2009) is used to attribute the characteristic of intra- and inter-annual variations in streamflow based on simulated daily discharge for baseline period and future periods (2030-2059 and 2070-2099) under the three different WUS scenarios (No-, Constant-, and Projected-WUS). An ANOVA test is applied with a significance level of 5% (p-value = 0.05) to evaluate the significant differences of IHA in near and far future of each of the aforementioned scenarios compared to the baseline period as suggested in Vu et al. (2019).

The Range of Variability Approach (RVA) established by Richter et al. (1997) is applied to evaluate streamflow regime alteration caused by climate change and groundwater withdrawals (WUSs). The RVA category thresholds are set as the median $\pm 25^{th}$ percentile of the models setup period data for each hydrologic indicator using non-parametric statistics. The degree of alteration (DA) is calculated as (The Nature Conservancy, 2009):

$$DAi = \frac{Roi - Rei}{Rei} * 100\%, \tag{2}$$

Where $DAi$ is the degree of hydrologic alteration of the $i^{th}$ IHA; $Roi$ and $Rei$ are the number of observed and expected repetitions in the scenario period for the $i^{th}$ IHA falling within the RVA target range. $Rei$ is defined as:

$$Rei = \gamma Rt, \tag{3}$$

Where $\gamma$ is the proportion of a single indicator's values falling within the RVA target range in the near and far future, i.e. $\gamma = 0.5$ is the suggested RVA target range between the $25^{th}$ and $75^{th}$ percentile values. $Rt$ is the total number of values for each indicator in the near and far future (30 years period), i.e. $Rt = 30$ (Richter et al., 1997; Zhang et al., 2019).

To evaluate the magnitude of change for each indicator, Richter et al., (1998) divided *DAi* (absolute value) into three classes: 0–±33% represents no or low alteration (L), ±33%–±67% represents moderate alteration (M), and ±67%–±100% represents high alteration (H). Positive RVA values indicate that the indicator remains stable within the upper and lower bounds (RVA targets) and negative RVA indicates, where the indicator is moving outside the upper or lower bounds to an alternative state.

## 3 Results

### 3.1 Groundwater sustainability

Groundwater sustainability assessment is evaluated on the sub-basin and entire basin scale.

### 3.1.1 Sub-basin scale

The SWAT model of the Halilrood Basin is divided into 285 sub-basins, however, WUSs are located only in 73 sub-basins corresponding to almost 33% (around 2385 km$^2$) of the total area of the Halilrood Basin. 31 of all 73 sub-basins with WUSs are in a sustainable state (groundwater recharge (GWR) > groundwater demand (GWD)) in the baseline period, however, in 42 sub-basins (17% of the total area) the GWD is higher than GWR. Less than 50% of water demand can be sustainably withdrawn from the groundwater in 22 sub-basins and less than 20% in 8 sub-basins.

The impact of climate change on GWR is assessed in the future periods for Constant-WUS scenario (Figure 3b and d). In the near future (Figure 3b), the number of sub-basins with a sustainable state (GWR > GWD) decreases from 31 (baseline period) to 26, while the unsustainable sub-basins (GWR < GWD) covering an area of 1211 km$^2$ (baseline period) increases to 1419 km$^2$ (20% of the total area). In the far future (Figure 3d), 25% of the entire basin (55 sub-basins) reach an unsustainable state, where less than 50% of water demand can be sustainably provided by groundwater in 24 sub-basins and among these, 9 sub-basins can only provide 20% of the water demand.

As shown in Figure 3c and e, where the two stressors climate change and growing water demand are considered simultaneously (Projected-WUS), supplying water sustainably is becoming more difficult in the near and far future when compared to the baseline period. Already 25% of the entire basin reach an unsustainable state in the near future (Figure 3c), similar to what is estimated to occur in the far future under the Constant-WUS scenario (Figure 3d). In the far future, among 73 sub-basin with WUSs, only 8 sub-basins are sustainable and in 56 sub-basins groundwater only provides less than 50% of the water demand (Figure 3e). Among these 56 unsustainable sub-basins, groundwater can only satisfy 20% of the water demand in a majority of 42 sub-basins.

### 3.1.2 Entire basin scale

GWR is simulated for the baseline, near, and far future periods (Table 7). The GWR is estimated to decrease under future climate change. This reduction is more severe in the far future, when it drops from 385 Mm$^3$ yr$^{-1}$ in the baseline period to 172 Mm$^3$ yr$^{-1}$. The currently sustainable groundwater situation for the entire Halilrood Basin (total GWD is lower than total GWR)

is expected to remain sustainable under future climate conditions, if we only account for the minimum and maximum water consumption for the growing population (min- and max-consumption/GWR < 1). However, if we consider the future increases in the number of WUSs (Projected-WUS), groundwater is only sustainable in the near future (GWR/GWD > 100%), whereas in the far future GWR is only able to fulfil 75% of the total demand (Table 8).

## 3.2 Streamflow sustainability

The alterations in each hydrologic indicator under future climate conditions (median, min, and max climate models) and different WUS scenarios are shown in Figure 4.

### 3.2.1 IHA-Group 1

The median monthly streamflows are expected to decrease in the future. This reduction is not significant for all indicators in the near future under No-WUS, although, a moderate RVA change is shown in late spring, summer, and early autumn. Three out of 12 and eight out of 12 indicators are significantly altered in scenarios Constant-WUS and Projected-WUS, respectively (Table 9).

In the far future, in eleven out of 12 months, median streamflows are expected to decrease significantly (Table 9), and the streamflow changes in Aug, Sep, and Oct are classified as "high" for all scenarios.

Strongest changes in monthly streamflow are expected for March under the Projected-WUS scenario where the streamflow decreased by 13.2 and 20.2 $m^3$ $s^{-1}$ respectively in the near and far future (Table 9). This might be due to the higher reduction in projected winter precipitation compared to the observations (Mahmoodi et al., 2020b).

The magnitude of changes expected under the three WUS scenarios (No-, Constant-, and Projected-WUS) are different. For instance, for the month of March which is subject to the strongest impact, the expected decrease under No-WUS scenario (corresponding to the singular impact of climate change) is 10.1 $m^3$ $s^{-1}$ in the near future, whereas under Constant- and Projected-WUS scenarios (corresponding to the impact of climate change and growing groundwater demand) the expected decreases are 11.6 $m^3$ $s^{-1}$ and 13.2 $m^3$ $s^{-1}$, respectively.

The uncertainty range of alterations in monthly streamflow under the min and max climate models indicate that uncertainty associated with the climate projections is higher in summer, fall, and winter seasons compared to spring season (i.e. April to June) when the degree of alteration varies between -13 to -100. This shows that the climate models consistently predict future spring streamflows outside the current 25th and 75th percentiles. Moreover, the direction of changes in spring season remains constant under different climate conditions projected by different climate models.

### 3.2.2 IHA-Group 2

In the near future, none of minimum streamflow indicators is expected to change significantly for No-WUS and Constant-WUS, while three out of five indicators will decrease pronouncedly for Projected-WUS (Table 9). In the far future scenario, the alteration in all minimum streamflow indicators is classified as "high" and decreases significantly, as the seasonal moving

average declines by 1.1 m$^3$ s$^{-1}$ (87%) under the three scenarios (No-, Constant-, and Projected-WUS; Table 9). Although annual extreme streamflows mainly experience a lower degree of change in the near and far future, the change is more significant for all indicators in the far future for the Projected-WUS scenario for which seasonal maximum streamflow decrease 22.4 m$^3$ s$^{-1}$ (35%) compared to the baseline period (Table 9). Also, alteration in the magnitude of base flow is estimated to be moderate and high in the near and far future, respectively. However, this alteration is only significant when WUS are considered. The reduction of base flow during the near future under climate change is 0.01 m$^3$ s$^{-1}$ (44%), which doubles when both climate change and extraction are considered in the future simulation (Table 9).

A wide uncertainty range of alterations (from -100 to +87 %) exists for the low-flow indicators i.e. 1 day min, 3 day min, 7 day min, and 30 day min. This indicates that the direction of alteration is associated with high uncertainties for the lowest streamflow and base flow indicator. The uncertainty is lower for the 30- and 90-day low flow values. In contrast, the annual extreme high streamflow indicators (e.g., 90 day max) consistently move outside the RVA target range, which is predicted for both the min and max climate models.

### 3.2.3 IHA-Group 3

Lowest streamflows are projected to occur earlier in all three scenarios, around three months for the near future (shift from Sep to June) and more than four months for the far future (shift from Sep to April). Also, the date of peak streamflow will shift by around two months and is estimated to happen earlier (shift from March to January) in both the near and far future of all scenarios. The uncertainty range shows that the alteration caused by different climate model projections is more pronounced for the time of occurrence of high flows, as the percentage of alteration varies from +47 to -53, compared to the occurrence of low flows with positive alteration (between +13 and +100) under different climate projections.

### 3.2.4 IHA-Group 4

The number of low streamflow pulses is estimated to increase in the future but this change is not significant in any scenario. The duration of low streamflow pulse is expected to increase significantly in the near future for all scenarios, whereas it is not significant in the far future except for the Projected-WUS scenario. The number of high streamflow pulses decreases significantly only in the near future for Projected-WUS scenario. The duration of high streamflow pulses does not change significantly in the near and far future in all scenarios. Number of days with no streamflow will increase significantly in both the near and far future under the three scenarios. This alteration is more severe for the far future under Projected-WUS scenario with 136 days more no-flow days as compared to the baseline period (Table 9).

The alterations in frequency and duration of high and low streamflow pulses under No-WUS in the near and far future, are similar to the alterations expected under Constant- and Projected-WUS. For instance, the number of high pulses (Hi Pulse) is estimated to reduce similarly (-2) under all three scenarios. Frequency and duration of high and low pulses do not change under the full range of climate projections and, with the relatively narrow uncertainty band, can therefore be assessed as robust projections.

### 3.2.5 IHA-Group 5

The number of fall and rise rates in streamflow are subject to significant changes only in the far future under No- and Constant-WUS scenarios (Table 9). The alteration for these indicators lies in the lower range for the median model. The full range of climate impacts causes a high degree of alteration in fall and rise rates (-47% to +93%).

The Non-parametric IHA scorecard is displayed in the supplementary material (Table S1). This shows a comparison of statistics (e.g., the low and high streamflow thresholds and annual coefficient of variation) for the baseline period and the future period. Moreover, the annual values and total distribution of each hydrologic parameter for the baseline period and two future periods under different WUS scenarios are shown in the supplementary material (Figure S1 and S2).

## 4 Discussion

The spatio-temporal variations of the groundwater demand to groundwater recharge ratio in the Halilrood Basin are compromising groundwater sustainability in the near and far future. These challenges are expected to be more severe when both climate change and population growth are considered. In addition, groundwater sustainability on the sub-basin scale for the Projected-WUS scenario as compared to Constant-WUS shows that the increases in groundwater demand and consumption exacerbate the negative impact of climate change on groundwater sustainability. To predict future groundwater demand, we used population growth as the main driver. However, increases in number of days with zero streamflow coincide with higher temperature and evaporation rate, and shifts in the precipitation regimes caused by climate change (Mahmoodi et al., 2020b). While this reduction is considered in water availability, the changing climate may lead to increasing irrigation requirements and may put the existing water use systems under additional pressure as similarly revealed in Toews and Allen (2009).

The rising water demand and WUSs will cause a decline of groundwater levels, due to the imbalance between the groundwater recharge under climate change and estimated groundwater demand in the future. This is not only resulting in an unsustainable groundwater use on sub-basin level and in the entire basin, but also changes the hydrologic regime and ecosystem condition by reducing the contribution of groundwater to streamflow, as 22 and 27 indicators show significant changes respectively for the near and far future under the Projected-WUS scenario. This is in agreement with findings by Haghighi et al. (2020) who stated robust changes in low streamflow indicators of Marboreh Basin in western Iran under future climate conditions.

The evaluation of indicators defined for monthly streamflows in the near future show that growing groundwater demand strongly affects the hydrologic regime of the Halilrood Basin during the dry season (spring, summer, and autumn) as opposed to the wet season (winter), when the changes of monthly streamflows are not significant under the Projected-WUS scenario. This is in agreement with the findings of Kakaei et al., (2018) which revealed substantial deficits in river discharge during the dry season (summer) of the Eskandari Watershed in central Iran due to human activities (abstraction of groundwater and surface water for irrigation purpose).

The predicted unsustainability of groundwater use could be even more intense if we focus on the changes projected for the magnitude and timing of annual extreme conditions, in which base flows, minimum and maximum streamflows are projected

to decrease and a four months shift is expected for minimum streamflows from Sep to June. This could lead to a higher groundwater demand during summer when surface water does not meet the rising demand, which is different in other seasons. In the near and far future, monthly streamflows and annual extreme streamflows are expected to decrease. However, the different magnitude of changes under the three WUS scenarios (No-, Constant-, and Projected-WUS) indicated that the influence of climate change on the streamflow regime alteration is stronger than growing groundwater demand. This is in agreement with previous studies e.g., Döll and Zhang (2010) and Shahid et al. (2018). In addition, the similar results for timing, frequency and duration of extreme hydrologic parameters under all three scenarios also showed that their changes are mainly caused by climate change.

Since the Halilrood River is the most important source of water in the region, the significant changes in hydrologic alteration indicators may have an impact on the ecosystem of the Wadi and Jazmorian wetland (water presence, area of water body, water depth, and wetland species). We are expecting smaller inundated area and shallower water body in Jazmorian wetland under climate change condition and groundwater withdrawal, as 27 hydrologic regime indicators show substantial alterations since out of 32 RVA 12 are classified as "high" and 15 as "moderate". Simultaneously, the availability of water for the wetland is reduced since, among 23 IHA considered for the magnitude of monthly streamflows and annual extreme streamflows, 21 IHA indicate significant changes and 15 IHA show high and moderate levels of alteration based on the RVA approach. Moreover, we expect lower water availability in future for the wetland due to increases in the number and duration of low pulses and number of days with zero streamflow as well as decreases estimated for the number and duration of high pulses. The significant alteration in falling rates, coinciding with alteration in the magnitude of streamflows, might influence soil moisture in the wetland and consequently change the distribution of the plants by an intensification of drought stress on plants, preventing wind and water erosion in the Jazmorian wetland. In summary, hydrologic regime alteration caused by climate change and growing groundwater demand, will contribute substantially to the ecological change of the wetland and hence, influence the freshwater ecosystem of Wadis in central Iran according to our RVA analysis.

Assessing the streamflow regime changes using IHA in conjunction with RVA, provide a proxy on initial ecological responses to the hydrologic regime changes without having to explicitly investigate ecological indices or building ecological models. However, in order to understand detailed ecological consequences and to identify hydrological thresholds for sustaining the complete or parts of the wetland ecosystem, an in-depth study involving ecological indicators and species requirements is nevertheless needed. The RVA approach enables researchers to link and track the hydrologic and ecological responses to the desirable implementations or ecosystem research efforts. Since, the RVA targets were set as the median $\pm 25^{th}$ percentile of the baseline period data for each hydrologic indicator, the high variation of the streamflow data in Wadi systems might lead to a high range of RVA targets. Therefore, we recommend a combination of RVA approach and a statistical method such as ANOVA to test the level of alteration and their significance in different hydrologic indicators.

The range of alteration derived from the min and max climate model projections allows to investigate how the climate models contribute to the uncertainty in projected hydrological changes. The derived uncertainties vary across the hydrological indicators. For instance, the magnitude of extreme streamflow events are highly uncertain for the low streamflow events as

opposed to lower uncertainty shown for the high streamflow events. Similarly, Cui et al., (2018) found that the uncertainty for low-flow periods under different climate projections is higher than for high-flow periods in the Yellow river, China. Projections of streamflow alterations in April, May, and June are more robust as opposed to other month of the years, as the uncertainty band of the min and max climate models is relatively narrow. The streamflow for these months is mainly generated from lateral

flows and snowmelt, which are both expected to change under the projected seasonal temperature increases for all climate models (Mahmoodi et al., 2020b). Temperature increases can cause a transformation in the pattern and type of precipitation, leading to more rain than snow, which is also reported for other arid regions in Iran (e.g., Shahvari et al., 2019). The lower RVA target (25[th] percentile of the baseline) for the magnitude of low flow extreme events (1-7 day min) and for the base flow indicator is zero and 0.007 respectively. Therefore, these indicators cannot be significantly reduced further and future changes

are likely to occur only under wetter climate conditions. The number and duration of low flow pules shows a strong alteration regardless of which climate model is used, which is likely driven by the reduction of groundwater contribution to streamflow under all possible future climate conditions (Mahmoodi et al., 2020b). When considering the uncertainty originating from the climate models, it is unrealistic to expect more optimistic conditions for the already threatened Jazmorian wetland. For instance, the degree of alteration and reduction in the magnitude and duration of high streamflow pulses remains constant even

under the wettest climate conditions in the future (max climate model). The alterations for different indicators under the median climate model is always within the uncertainty band, while for some indicators the alterations approach the upper or lower bound. This can be explained by the selection method of the median, min and max climate models, which is carried out based on the lumped water balance components and not the individual indicators.

**5 Conclusions**

The spatio-temporal variation of groundwater sustainability and the streamflow alteration in the near and far climate change-impacted future have been assessed under five different scenarios: (i) no groundwater demand (ii) unaltered present-day groundwater demand (iii) an increase in groundwater demand (iv) minimum-, and (v) maximum water consumption. Our findings show that:

1) The significant reduction estimated for groundwater recharge under climate change coincides with rising demand from

WUSs and water consumption.

2) The growing groundwater demand in the future exacerbates the impact of climate change on the sustainable use of water resources in the Halilrood Basin.

3) A sustainable state is possible for the entire Halilrood Basin in near and far future if only consumptive water use is considered. However, several sub-basins would still be extremely unsustainable. Hence, water provisioning from sustainable

to unsustainable sub-basins would be required.

4) The impacts of climate change and growing groundwater demand on the freshwater ecosystems in the Jazmorian wetland basin are expected to be intensified as considerable hydrologic regime alterations projected in the Halilrood River (27 IHA

indicators show significant changes in the far future and among these the RVA is classified as "high" and "moderate" for 18 IHA).

5) Uncertainties originating from the climate model ensemble are higher for the monthly streamflow in summer, fall, and winter season and extreme low flows compared to the streamflow of spring season and the number and duration of low streamflow pulse indicators.

The combined results show that climate change has a stronger impact on hydrologic regime alterations and consequently on the freshwater ecosystem in the near and far future as compared to growing groundwater demand in Halilrood Basin. The

presented results are useful for long-term planning which is required for a sustainable water resources management under changing future conditions.

## Author contribution

NM, JK, and PDW developed the study design. NM performed the model simulations with essential support of JK who carried out the EURO-CORDEX data processing. All authors substantially contributed to the interpretation of the results. NM wrote

the initial draft of the manuscript and all co-authors discussed the results and revised the work carefully.

## Competing interests

The authors declare that they have no conflict of interest.

## Acknowledgements

The German Academic Exchange Service (DAAD) and the Federal Ministry of Education and Research (BMBF) granted this

research, through the special program 'Sustainable Water Management'(NaWaM/-ID: 57260501). JK acknowledges funding through the "GLANCE" project (Global change effects in river ecosystems; 01LN1320A) supported by BMBF.

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

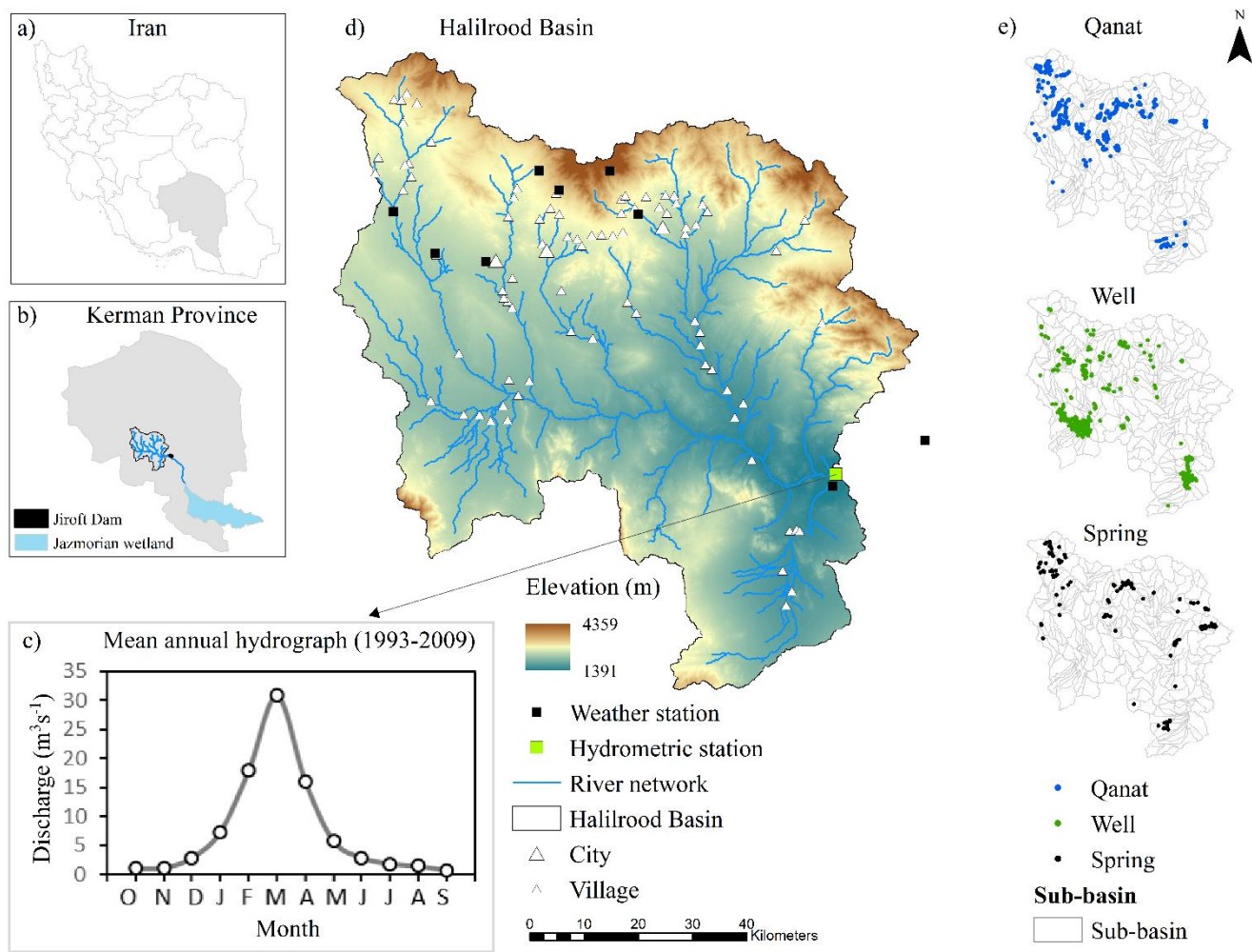


**Figure 1: Location of the Halilrood Basin, water use systems, and monitoring stations considered in this study. Average monthly streamflows derived from the observed data at the outlet of the basin.**

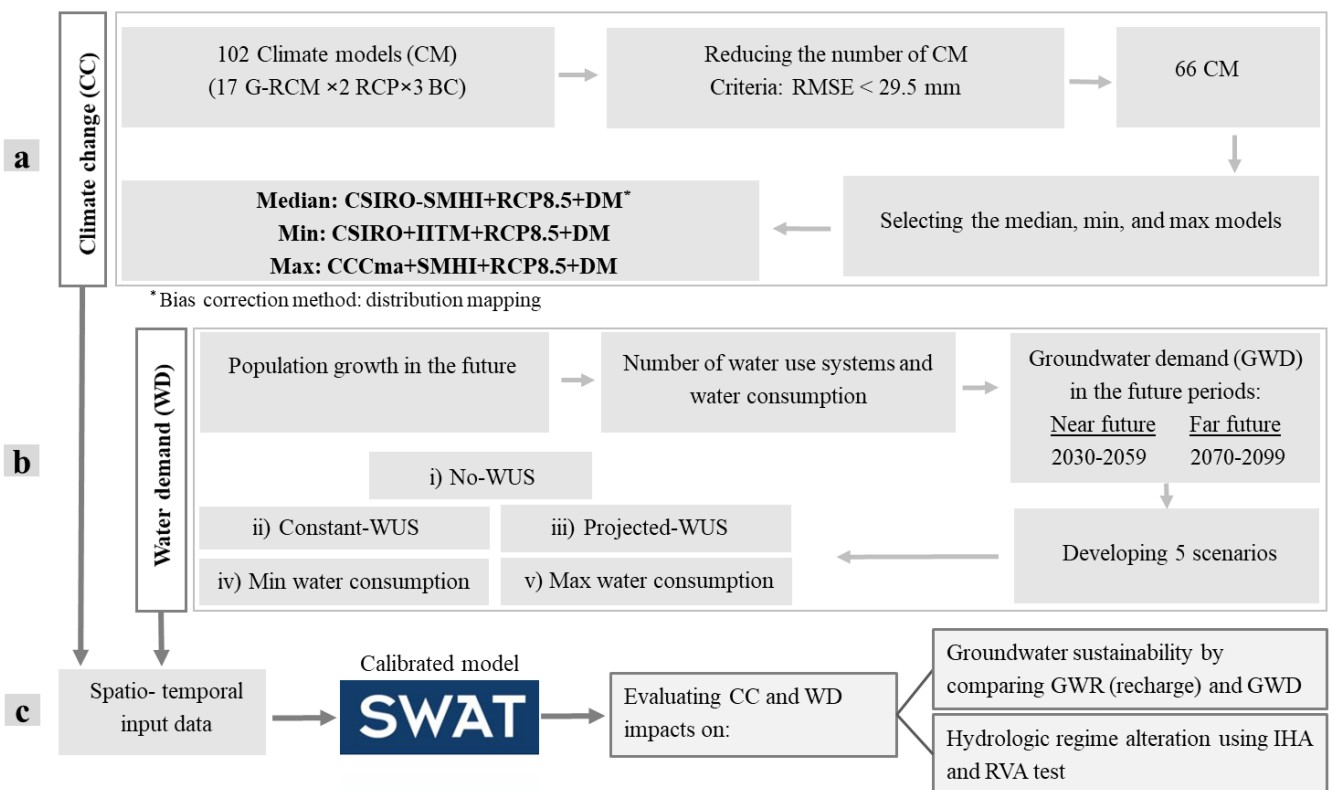

**Figure 2: Flow chart of the methodology employed.**

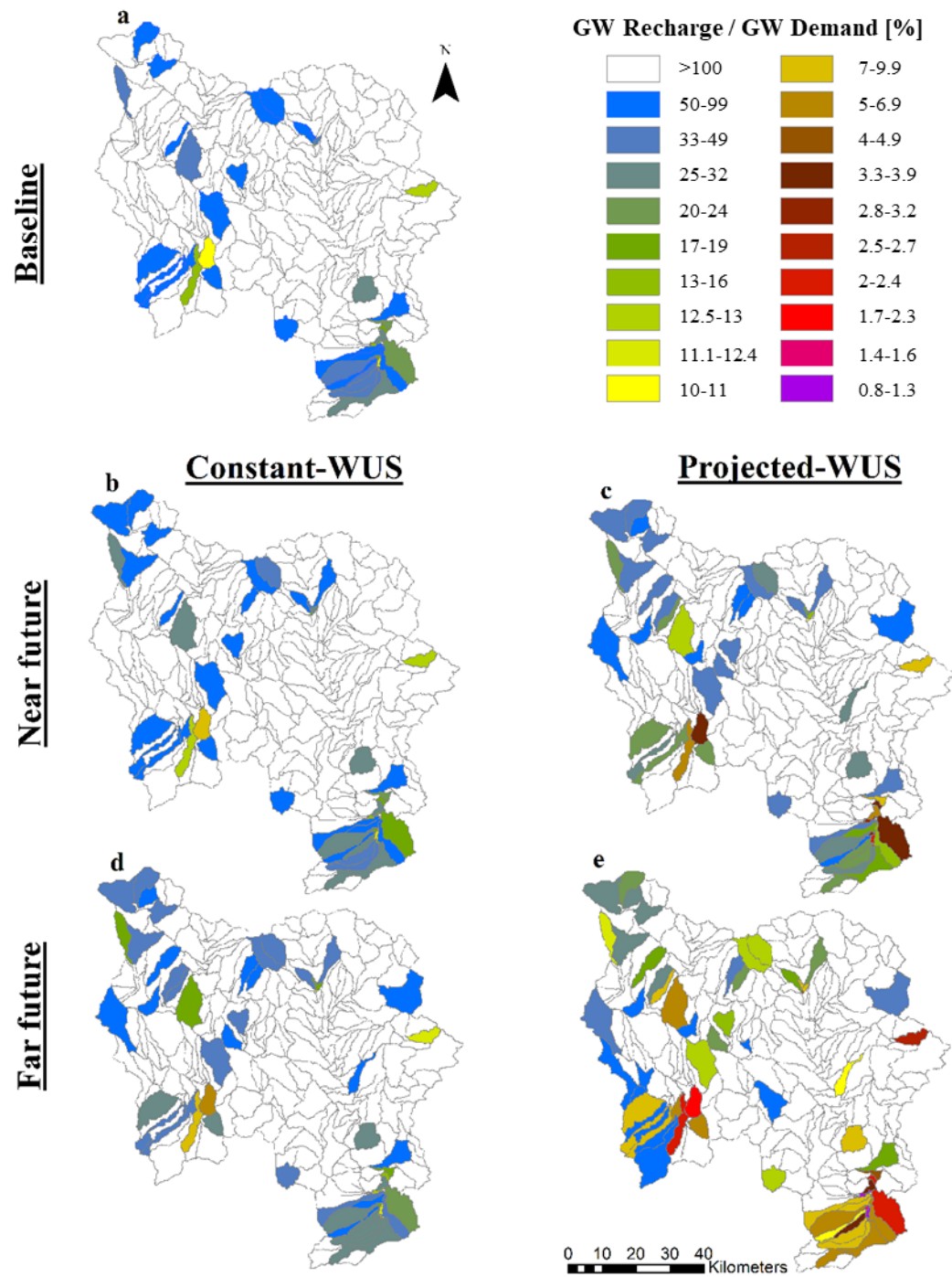


**Figure 3: The percentage of groundwater demand that can be sustainably met by groundwater recharge at sub-basin scale under two different scenarios: Constant-WUS: the number of water use systems in the basin remain unaltered in the future, and Projected-WUS: the number of water use systems increase linearly with population growth.**

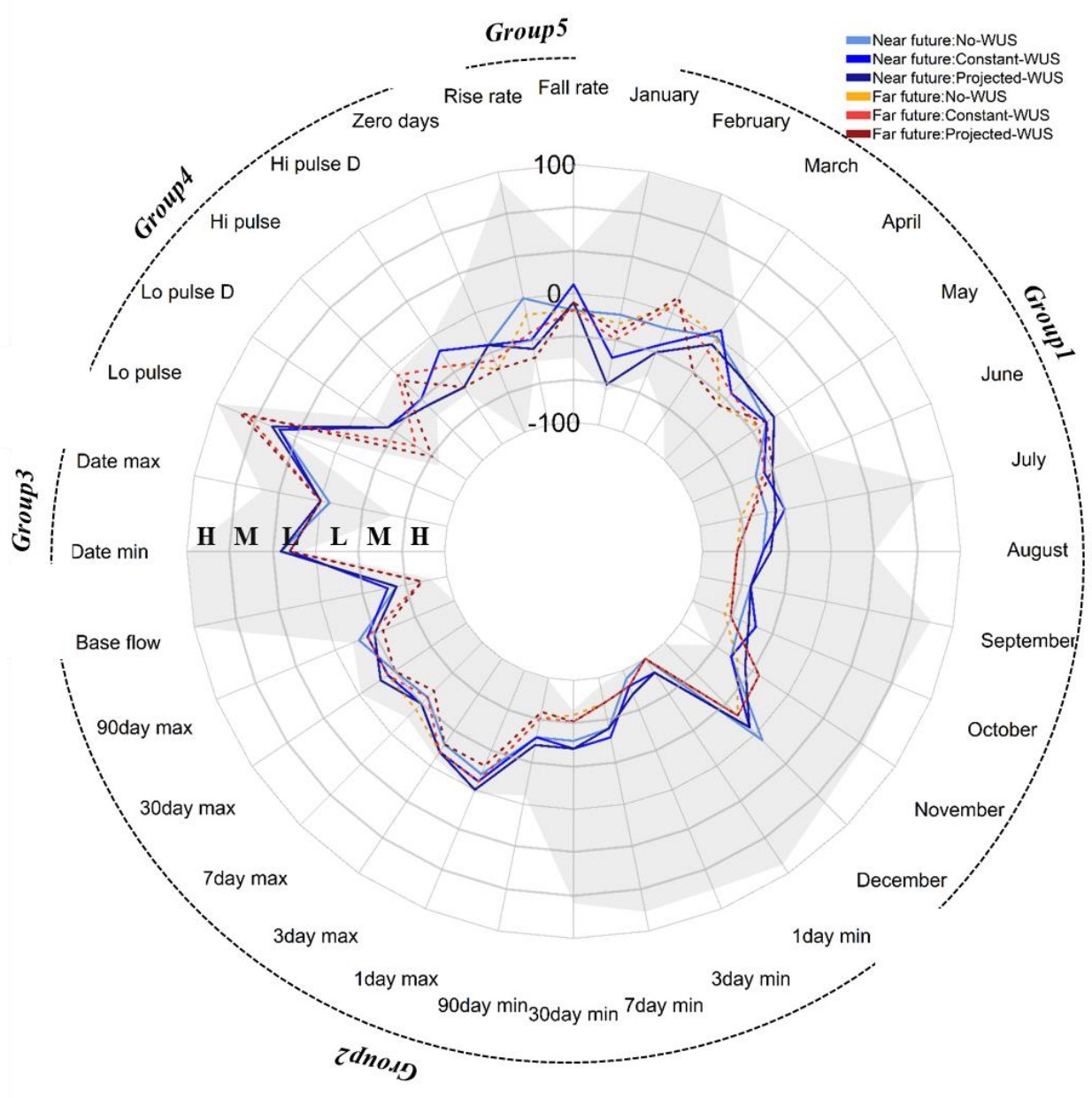

Figure 4: RVA (The Range of Variability Approach) deviation and classes of alteration (High (H), Moderate (M), and Low (L)) for each IHA (Indicators of Hydrologic Alteration) indicator in the near and far future under three different scenarios: No-WUS: the water use systems are not considered, Constant-WUS: the number of water use systems in the basin remain unaltered in the future, and Projected-WUS: the number of water use systems increase linearly with population growth. Alteration (uncertainty) band derived from min and max climate models for both near and far future simulations under three WUSs scenarios is shown in grey.

**Table 1: Statistical analysis of annual streamflow (m$^3$ s$^{-1}$) simulated for the baseline (1979-2009) and two future periods (2030-2059 and 2070-2099) for the median, min (driest) and max (wettest) climate models.**

| | | | Median climate model | | Max climate model | | Min climate model | |
|---|---|---|---|---|---|---|---|---|
| | Observations | Baseline | Near future | Far future | Near future | Far future | Near future | Far future |
| | (1993-2009) | (1979-2009) | (2030-2059) | (2070-2099) | (2030-2059) | (2070-2099) | (2030-2059) | (2070-2099) |
| **Mean** | 7.66 | 13.31 | 9.93 | 5.84 | 11.62 | 15.11 | 2.85 | 2.13 |
| **Max** | 33.21 | 39.79 | 38.75 | 20.78 | 48.59 | 76.97 | 6.74 | 5.72 |
| **Min** | 0.43 | 1.56 | 0.53 | 0.85 | 0.77 | 0.3 | 0.34 | 0.21 |
| **Median** | 3.42 | 11.73 | 6.67 | 3.74 | 5.84 | 10.34 | 2.34 | 1.51 |
| **STDEV** | 8.20 | 10.27 | 9.77 | 5.34 | 13.58 | 17.20 | 1.72 | 1.48 |
| **SKEW** | 1.88 | 0.98 | 1.50 | 1.55 | 1.52 | 1.94 | 0.76 | 0.99 |

**Table 2: Population of cities located in the Halilrood Basin according to the last census in 2017 and the future population projected based on the population growth rate suggested by PBO (2019).**

| Cities | Population 2011 | Population 2017 | Mid. of near future 2045 | Mid. of far future 2085 |
|---|---|---|---|---|
| **Bazanjan** | 4325 | 4517 | 5592 | 7127 |
| **Baft** | 80528 | 84103 | 104119 | 132714 |
| **Rabor** | 33859 | 35362 | 43778 | 55801 |
| **Total basin** | 118712 | 123982 | 153489 | 195643 |

**Table 3: Projected water demand from water use systems. Number of water use systems mentioned in parenthesis are estimated based on population growth in 2017 and in the near and far future.**

| WUS | Mean discharge (m³ s⁻¹) | Year 2011-reported (Mm³ yr⁻¹) | Year 2017 (Mm³ yr⁻¹) | Mid. of near future-2045 (Mm³ yr⁻¹) | Mid. of far future-2085 (Mm³ yr⁻¹) |
|---|---|---|---|---|---|
| **Well** | 0.01152 | (329) 119.52 | (344) 124.83 | (425) 154.54 | (542) 196.98 |
| **Qanat** | 0.00211 | (262) 17.43 | (274) 18.21 | (338) 22.54 | (431) 28.73 |
| **Spring** | 0.00134 | (170) 7.16 | (170) 7.16 | (170) 7.16 | (170) 7.16 |
| **WUS** | 0.01497 | (761) 144.12 | (787) 150.20 | (934) 184.24 | (1134) 232.87 |

M: million

**Table 4: Minimum and maximum estimated demand for consumptive water use according to the data reported for the water required and population growth currently and in the future.**

| | Year 2011-reported (Mm$^3$ yr$^{-1}$) | Year 2017 (Mm$^3$ yr$^{-1}$) | Mid. of near future-2045 (Mm$^3$ yr$^{-1}$) | Mid. of far future-2085 (Mm$^3$ yr$^{-1}$) |
|---|---|---|---|---|
| **Min. water consumption: 0.135 m$^3$ day$^{-1}$ person$^{-1}$** | 5.84 | 6.11 | 7.56 | 9.64 |
| **Max. water consumption: 0.300 m$^3$ day$^{-1}$ person$^{-1}$** | 12.99 | 13.58 | 16.8 | 21.42 |

**M: million**

**Table 5: Scenarios included in near and far future simulations to evaluate groundwater sustainability and hydrologic regime alteration on different spatial scales.**

| Scenarios | Description | Climate change (median, min, and max) | WUSs including water consumption | Water consumption only | Groundwater sustainability | | Hydrologic regime change |
|---|---|---|---|---|---|---|---|
| | | | | | Sub-basin scale | Entire basin scale | |
| **i. No WUS** | Water use systems do not exist | ✓ | | | | | ✓ |
| **ii. Constant-WUS** | Currently existing water use systems in the basin remain unaltered | ✓ | ✓ | | ✓ | | ✓ |
| **iii. Projected-WUS** | The number of water use systems increase linearly with population growth | ✓ | ✓ | | ✓ | ✓ | ✓ |
| **iv. Min-consumption** | Minimum amount of water required per person per day in Iran | ✓ | | ✓ | | ✓ | |
| **v. Max-consumption** | Maximum amount of water required per person per day in Iran | ✓ | | ✓ | | ✓ | |

✓ Addresses the scenario/s considered for each analysis

**Table 6: The used set of 32 indicators of hydrologic alteration categorized into five groups (Richter et al., 1997).**

| IHA parameters group | Hydrologic parameters | Unit |
|---|---|---|
| **Group 1. Magnitude of monthly water conditions** | Median flow for each calendar month | $m^3 s^{-1}$ <br> $m^3 s^{-1}$ |
| **Group 2. Magnitude of annual extreme discharge events with different durations** | 1-day minimum flow (1-day min) <br> 3-day minimum flow (3-day min) <br> 7-day minimum flow (7-day min) <br> 30-day minimum flow (30-day min) <br> 90-day minimum flow (90-day min) <br> 1-day maximum flow (1-day max) <br> 3-day maximum flow (3-day max) <br> 7-day maximum flow (7-day max) <br> 30-day maximum flow (30-day max) <br> 90-day maximum flow (90-day max) <br> Base flow index (Base flow) | $m^3 s^{-1}$ <br> $m^3 s^{-1}$ <br> $m^3 s^{-1}$ <br> $m^3 s^{-1}$ <br> $m^3 s^{-1}$ <br> $m^3 s^{-1}$ <br> $m^3 s^{-1}$ <br> $m^3 s^{-1}$ <br> $m^3 s^{-1}$ <br> $m^3 s^{-1}$ <br> $m^3 s^{-1}$ |
| **Group 3. Timing of annual extreme water conditions** | Date of annual minimum flow (Date min) <br> Date of annual maximum flow (Date max) | day of year <br> day of year |
| **Group 4. Frequency and duration of high and low pulses** | Number of low pulses each year (Lo pulse) <br> Number of high pulses each year (Hi pulse) <br> Duration of low pulses (Lo pulse D) <br> Duration of high pulses (Hi pulse D) <br> Number of zero flow days (Zero days) | dimensionless <br> dimensionless <br> dimensionless <br> dimensionless <br> days |
| **Group 5. Rate and frequency of water condition changes** | Median rate of positive changes in flow (Rise rate) <br> Median rate of negative changes in flow (Fall rate) | $m^3 s^{-1} day^{-1}$ <br> $m^3 s^{-1} day^{-1}$ |

**Table 7: Average annual groundwater recharge simulated on the entire basin scale in different periods.**

| | Baseline (1979-2009) | Near future climate scenario (2030-2059) | Far future climate scenario (2070-2099) |
|---|---|---|---|
| **Groundwater Recharge (Mm$^3$ yr$^{-1}$)** | 385 | 311 | 172 |

M: million

**Table 8: Groundwater sustainability on the entire basin scale under three scenarios: Projected-WUS: the number of water use systems increase linearly with population growth, min- and max consumption: the minimum and maximum water demand corresponded to population growth in the future. The percentage of the water demand that can be sustainably provided by groundwater is mentioned in brackets.**

| Scenarios | Groundwater sustainability | | |
|---|---|---|---|
| | Baseline (1979-2009) | Near future (2030-2059) | Far future (2070-2099) |
| **Projected-WUS** | 0.4 (250%) | 0.59 (170%) | 1.35 (75%) |
| **Min-consumption** | 0.015 (6600%) | 0.024 (4200%) | 0.056 (1700%) |
| **Max-consumption** | 0.034 (2900%) | 0.054 (1850%) | 0.124 (800%) |

**Table 9: Absolute change for each Indicators of Hydrologic Alteration (IHA) (significant changes highlighted in bold digits) in the future under three different scenarios: No-WUS: the water use systems are not considered, Constant-WUS: the number of water use systems in the basin remain unaltered in the future, and Projected-WUS: the number of water use systems increase linearly with population growth. Percentage of RVA (The Range of Variability Approach) deviation is shown in brackets.**

| IHA groups | IHA | Near future (2030-2059) | | | Far future (2070-2099) | | |
|---|---|---|---|---|---|---|---|
| | | NO-WUS | Constant-WUS | Projected-WUS | NO-WUS | Constant-WUS | Projected-WUS |
| Group1 | January | -6.3(-13) | -7.0(-47) | -7.8(-68) | **-8.8(-20)** | **-9.4(-33)** | **-9.9(-27)** |
| | February | -2.8(-13) | -4.2(-27) | -5.8(-33) | -9.6(+7) | -10.7(+7) | -11.9(+13) |
| | March | -10.1(0) | -11.6(+6) | -13.2(-7) | **-18.2(0)** | **-19.2(-13)** | **-20.2(-33)** |
| | April | -5.5(-13) | **-6.4(-27)** | **-7.4(-13)** | **-10.1(-40)** | **-10.6(-27)** | **-11.1(-40)** |
| | May | -2.5(-20) | -3.2(-20) | **-3.8(-13)** | **-5.3(-27)** | **-5.6(-27)** | **-5.9(-20)** |
| | June | -1.4(-47) | -1.8(-40) | -2.2(-33) | **-3.3(-40)** | **-3.5(-40)** | **-3.7(-33)** |
| | July | -1.1 (-47) | -1.4(-33) | **-1.7(-40)** | **-2.4(-68)** | **-2.5(-60)** | **-2.6(-60)** |
| | August | -1.1(-53) | -1.2(-53) | **-1.4(-47)** | **-1.9(-73)** | **-2.0(-73)** | **-2.0(-73)** |
| | September | -0.7(-60) | -0.8(-60) | **-1.0(-60)** | **-1.3(-73)** | **-1.4(-73)** | **-1.4(-73)** |
| | October | -0.7(-60) | **-0.8(-47)** | **-0.9(-53)** | **-1.2(-73)** | **-1.3(-68)** | **-1.3(-68)** |
| | November | -0.5(-53) | -0.6(-53) | **-0.7(-40)** | **-0.8(-47)** | **-0.9(-27)** | **-0.9(-27)** |
| | December | -2.7(+7) | **-3.0(-7)** | **-3.3(-7)** | **-3.8(-20)** | **-3.9(-20)** | **-4.1(-20)** |
| Group2 | 1-day min | -0.2(-100) | -0.2(-87) | -0.2(-87) | **-0.3(-100)** | **-0.3(-100)** | **-0.3(-100)** |
| | 3-day min | -0.2(-93) | -0.3(-87) | **-0.3(-80)** | **-0.4(-87)** | **-0.4(-87)** | **-0.4(-87)** |
| | 7-day min | -0.3(-60) | -0.3(-53) | **-0.4(-60)** | **-0.4(-80)** | **-0.4(-80)** | **-0.4(+80)** |
| | 30-day min | -0.2(-53) | -0.3(-47) | -0.4(-47) | **-0.6(-73)** | **-0.6(-68)** | **-0.7(-68)** |
| | 90-day min | -0.6(-53) | -0.7(-53) | **-0.8(-47)** | **-1.1(-68)** | **-1.1(-68)** | **-1.1(-73)** |
| | 1-day max | -110.1(-13) | **-122.8(-7)** | **-135.5(0)** | **-199.1(-7)** | **-206.7(-7)** | **-214.3(-20)** |
| | 3-day max | -62.5 (-20) | -71.1(-13) | **-79.7(-13)** | **-115.2(-13)** | **-120.9(-13)** | **-126.6(-20)** |
| | 7-day max | -40.8(-40) | **-46.5(-33)** | **-52.1(-33)** | **-72.3(-27)** | **-76.3(-40)** | **-80.3(-47)** |
| | 30-day max | -22.7(-33) | **-25.6(-27)** | **-28.4(-20)** | **-36.3(-27)** | **-38.4(-27)** | **-40.5(-33)** |
| | 90-day max | -10.6(-20) | -12.5(-27) | **-14.4(-33)** | **-19.7(-27)** | **-21.0(-27)** | **-22.4(-40)** |
| | Base flow | -0.01(-60) | -0.01(-53) | **-0.02(-60)** | **-0.02(-80)** | **-0.02(-80)** | **-0.02(-80)** |
| Group3 | Date min | **-85(+27)** | **-84.0(+20)** | **-85(+27)** | **-137(+20)** | **-136.0(+20)** | **-137(+20)** |
| | Date max | -53(-7) | -52.0(0) | **-53(0)** | **-61(0)** | **-60.0(0)** | **-61(0)** |
| Group4 | Lo pulse | 2.6(+47) | 2.6(+47) | 2.6(+53) | 2.4(+73) | 2.4(+73) | 2.1(+80) |
| | Lo pulse D | **2.4(-27)** | **2.3(-27)** | **2.3(-27)** | 2.0(-53) | 2.1(-53) | **2.4(-68)** |
| | Hi pulse | **-2.0(-33)** | **-2.0(-33)** | **-2.0(-40)** | -1.6(-7) | -1.6(-7) | **-1.8(-13)** |
| | Hi pulse D | -1.9(-13) | -1.9(-13) | -2.3(-47) | -1.3(-27) | -1.3(-27) | -1.6(-47) |
| | Zero days | **78(-27)** | **79(-27)** | **83(-27)** | **130(-47)** | **132(-40)** | **136(-47)** |
| Group5 | Rise rate | 0.16(0) | 0.003(-33) | -0.08(-40) | **0.89(-13)** | **0.6(-27)** | 0.53(-47) |
| | Fall rate | -0.16(-13) | -0.01(+7) | 0.1(-7) | **-0.45(-13)** | **-0.25(-13)** | -0.08(-7) |