# Peer review of "Spatially distributed impacts of climate change and groundwater demand on the water resources in a Wadi system"

_Hydrology and Earth System Sciences, 2020_

## Author Comment (AC1)

**Reply to Referee #1**

Dear Referee,
We are grateful for your time and constructive comments and suggestions on our manuscript. The provided comments have contributed substantially to improving the paper. Below, we provide a point-by-point response (in blue) to the comments (in black) and explain how we will address each of the reviewers' comments in the revised manuscript.

Sincerely,
Nariman Mahmoodi, Jens Kiesel, Paul D. Wagner, Nicola Fohrer

**GENERAL COMMENTS:**

The present paper is on "Sustainable use of water resources…". The topic/paper is interesting but needs a number of modifications before it can be accepted.

"Sustainable/ sustainability" is mentioned several times in the text (and the title). In my understanding sustainable water resources management is not simply supplying an ever growing demand (that's a really old, outdated approach from the last century) but an important part is water demand management. In the paper there is nothing about water demand management, but in one of their scenarios the authors simply interpolate past/observed water use/demand into the future.

We agree that "sustainability of water resources" is a complex field and requires proper definition. In this study, we consider water related sustainability within two different aspects: groundwater sustainability (groundwater recharge vs groundwater withdrawals) and surface water sustainability (the range of variability approach). Although we did comprehensive analyses on these two sustainability aspects under different water use systems, water consumption-, and climate change scenarios (sections: 3.1 Groundwater sustainability and 3.2 Hydrologic alteration), we acknowledge that a clear definition is lacking. We will change the subtitle of section 3.2 to "streamflow sustainability" (as suggested in the specific comments) and add a clear definition of sustainability in the revised version.

Overall, the wording (or definitions) needs to be reconsidered. The authors define "sustainable/ sustainability" as state where "GWR>GWW"; i.e. "groundwater recharge" is higher than "groundwater withdrawal". A state where GWW/GWR ratio is higher than "1" is considered "unsustainable". The authors consider GWR only; there is no information (to my understanding) on the groundwater level or volume. The authors assume that the whole water demand can be withdrawn, e.g. stating that "The rate of GWW to GWR is greater than 2..." (p. 7, lines 195/196). Without information on the groundwater level or volume, how can they say that the water volume demanded can be withdrawn? In my opinion the whole text needs to be changed (I will define demand on groundwater here as "GWD"): the authors can only calculate what volume can be "sustainably" withdrawn from groundwater. In cases where "GWD>GWR" water withdrawals

will become unsustainable, i.e. water demand on groundwater resources is larger than groundwater recharge, meaning the groundwater level or volume will decline. The (or one version for) correct wording would be that "Only 50% of the water demand can be sustainably withdrawn…" instead of "The rate of GWW to GWR is greater than 2..." (p. 7, lines 195/196). I think the whole text (and Tables/Figures) needs to be changed in this direction.

Thank you very much for your comment. Indeed, there is no data on groundwater level or volume. We believe that our definition of sustainability is in agreement with the view of the reviewer: Withdrawing the amount of groundwater recharge would be sustainable —going beyond this amount is unsustainable. As we implemented groundwater withdrawal in the model, we used it for our calculation, but we agree with the reviewer that the proposed way of relating it to percentage of groundwater demand that can be sustainably withdrawn is more precise. Beyond considering 'groundwater sustainability' only, please note that our definition of 'sustainability' includes the IHA and RVA analysis and the impact of the scenarios (including groundwater withdrawals) on the ecological flow regime (streamflow sustainability). We will thoroughly revise the manuscript in this regard.

It is not clear if the results (e.g. "GWW/GWR") represented (e.g. Fig. 2) a calculated on an annual basis and then averaged over the entire (30 years) period or if it is calculated for the entire (30 years) period. If it is calculated on an annual basis and averaged afterwards, how is groundwater storage, e.g. in a very wet year GWR may be (much) higher than GWD and increase availability (increased groundwater level or volume) in the next year, considered? Also, to my knowledge, there is no groundwater flow between sub-basins in SWAT. In reality a sub-basin (region) with high groundwater withdrawals, resulting in declining groundwater levels, may receive groundwater inflow from neighbouring sub-basins (regions) without groundwater withdrawals, i.e. high(er) groundwater levels.

In our assessment groundwater recharge was calculated for the entire 30-year periods. We will clarify this in the revised manuscript. Not being able to consider groundwater connection between neighboring sub-basins is not a significant disadvantage, because we want to explicitly show the exact location (sub-basin/regions) of imbalances between GWR and GWD in the study area. However, we discussed the connectivity of groundwater in our analysis when the whole basin was treated as an integrated system and the total groundwater recharge was compared to the total water consumptions/demands. The results are shown in Table 6 and 7.

The paper builds on work of Mahmoodi et al. (2020a) (referenced as Mahmoodi, N., Kiesel, J., Wagner, D.P., and Fohrer, N.: Water use systems and soil and water conservation methods in a hydrological model of an Iranian Wadi system. J. Arid Land, 12, 545–560, https://doi.org/10.1007/s40333-020-0125-3, 2020a.), describing the set-up and calibration/validation of a SWAT model to the Halilrood river basin. In a simplified way the water balance can be given as:

$SW(t) = SW(t-1) + PRECIP(t) - SR(t) - ET(t) - PERC(t) - SSF(t)$

where SW(t) is the soil water content at time step t, PRECIP is precipitation, SR is surface runoff, ET is evapotranspiration, PERC is percolation, and SSF is subsurface flow (with the last two variables describing groundwater-/subsurface part of the system, this part may be differentiated differently depending on the complexity of the approach/model). The important point is that there are some variables that are observed usually, i.e. PRECIP and river flow (combining SR and SSF), while for other variables seldom or only for very few locations observed values are available, e.g. ET and PERC. Assuming (for simplicity) constant SW and known PRECIP and river flow, potential and actual evapo(-transpi-)ration and PERC are unknown. Depending on the parametrisation of the model potential and actual evapo(-transpi-)ration and PERC can differ strongly. In a model calibrated/validated on stream flow only there is high uncertainty if potential and actual evapo(-transpi-)ration and PERC are simulated correctly - here either observed values, e.g. lysimeter measurements (point information), or from other sources, e.g. LAI or actual evapotranspiration from GLEAMS or MODIS (areal information) should be used to validate the model results. In the paper of Mahmoodi et al. (2020a) such a validation is not carried out and therefore I wonder how the authors can prove the reliability of their SWAT parametrisation. For instance in Table 4 (Selected parameters for calibration in the SWAT model) Mahmoodi et al. (2020a) give "EVRCH Reach evaporation adjustment factor from 0.5 to 0.8", to my understanding this means that the potential evaporation is reduce strongly by 20 to 50% (i.e. more water may percolate, become groundwater recharge)! As given in Mahmoodi et al. (2020a, p. 551):

"WYLD=SURFQ+LATQ+GWQ–TLOSS, (1)

where WYLD is the water yield (mm); SURFQ is the surface runoff (mm); LATQ is the lateral flow contribution to stream (mm); GWQ is the groundwater contribution to stream flow (mm); and TLOSS is the transmission losses (mm), i.e., water loss via transmission through the bed of the channels." Transmission losses in semi-arid environments/rivers can be very high, introducing another uncertain variable in the simulations.

We agree with the reviewer that the model was -as most hydrologic models- not validated for evapotranspiration. However, going beyond other parameterization approaches it was calibrated using a multi-metric approach that includes the comparison to flow duration curves. The study area is a data scarce region. We used the SWAT model as it has proven its ability to perform well under these conditions (Wagner et al. 2012) and even in ungauged catchments (Srinivasan et al., 2010). The spatially distributed parameters of SWAT derived from land use, soil, and slope maps provide an initial parameterization that has proven its use in various regions worldwide. To address the reviewers concerns we will compare the model to potential and actual ET in the catchment and to a mean recharge rate for Iran. As we focus on the long-term water balance, we believe that these aggregated values for the water balance equation along with the validation of the dynamics of streamflow and the proven ability of SWAT to model water fluxes provide confidence in our model results.

In SWAT2012, a high portion of streamflow volume in low flow periods could be lost due to overestimation of channel evaporation since the default reach evaporation adjustment factor is equal to 1 (Arnold et al., 2013; Nguyen et al., 2018). According to the SWAT manual (Arnold et al., 2013) and other studies (Nguyen et al., 2018; Muleta and Nicklow 2005), the EVRCH default setting of 1 represents the upper threshold of reach evaporation and hence may lead to overestimation of evaporation from reaches. The parameter was introduced specifically for arid regions to reduce reach evaporation (Arnold et al., 2013, see chapter 4, page 112). We will further address this issue in the water balance comparison (see first paragraph of this response).

Wadi bed infiltration has an important impact on streamflow and recharge rate, but depends on local conditions and the quantification of bed infiltration is challenging since many factors i.e., evaporation, sediment and deposit properties, volume and velocity of surface discharge, slope and widths of the reaches are involved (Weather et al., 2008; Neitsch et al., 2011). Over/underestimation of reach evaporation and transmission losses, in combination, can affect the SWAT simulated stream flow. From a good/satisfactory performance of our hydrological model in simulating streamflow and potential evapotranspiration we can only infer that bed infiltration is represented sufficiently well.

References:

Neitsch, S.L., Arnold, J.G., Kiniry, J.R. and Williams, J.R., 2011. Soil and water assessment tool theoretical documentation version 2009. Texas Water Resources Institute.

Arnold, J.G., Kiniry, J.R., Srinivasan, R., Williams, J.R., Haney, E.B., Neitsch, S.L. 2013. SWAT 2012 input/output documentation. Texas Water Resources Institute.

Nguyen, V.T., Dietrich, J., Uniyal, B., Tran, D.A. 2018. Verification and correction of the hydrologic routing in the soil and water assessment tool. Water, 10(10), p.1419.

Wagner, P.D., Fiener, P., Wilken, F., Kumar, S. and Schneider, K., 2012. Comparison and evaluation of spatial interpolation schemes for daily rainfall in data scarce regions. Journal of Hydrology, 464, pp.388-400.

Srinivasan, R., Zhang, X., Arnold, J.G., 2010. SWAT ungauged: hydrological budget and crop yield predictions in the upper Mississippi river basin. Trans. ASABE 53 (5), pp.1533–1546.

Muleta, M.K. and Nicklow, J.W., 2005. Sensitivity and uncertainty analysis coupled with automatic calibration for a distributed watershed model. Journal of hydrology, 306(1-4), pp.127-145.

Furthermore, the paper builds on work of Mahmoodi et al. (2020b) (referenced as "Mahmoodi, N., Wagner, D.P., Kiesel, J., and Fohrer, N.: modeling the impact of climate change on streamflow and major hydrological components of an Iranian Wadi system. Water Clim. Change, https://doi.org/10.2166/wcc.2020.098, 2020b."): "The G-RCM CSIRO-SMHI was chosen since it

represented the median model of the major hydrological components (Mahmoodi et al., 2020b)." The authors write that they follow the argumentation of Tebaldi and Knutti (2007) and Thober and Samaniego (2014). However, reading the paper of Tebaldi and Knutti (2007) I only find that they discuss using (weighted) averages of a number of climate models, based on the idea that the performance can be improved by averaging or combining results from multiple models. But this is that first a number of climate models are run and then the results are averaged (in the case of impact models the input from a number of climate models are used and then results of the impact models are averaged). In the study of Thober and Samaniego (2014) a number of (meteorological) indices are used to select regional climate models, i.e. reduce the number of impact model runs. In case such an analysis (to select G-RCM CSIRO-SMHI) was carried out in the present study this analysis and its results need to be described (e.g. in the Suppl. Material), otherwise the results are not replicable (I also need to mention that Thober and Samaniego (2014) reduce the number but they do not propose to use results of only ONE climate model). In the paper of Mahmoodi et al. (2020b) I don*t find any information that "G-RCM CSIRO-SMHI … represented the median model of the major hydrological components". The use of only one climate scenario restricts the value of the paper – for a more wet or dry (climate) future results could be very different. Such results would be needed to enable more robust decisions on future water resources management, especially as the authors state that effects of climate change on surface/groundwater resources are much more significant than future water demand/use.

We believe that the focus on one climate model is instructive to discuss possible future impacts and the interaction of climate change impacts with the impacts of growing groundwater demand. The procedure of analysing the impacts of all climate models in an ensemble on the target indicator (here: streamflow) and then selecting the median model is common in climate change impact studies (ensemble of opportunity, model democracy). We therefore determined a median model using the following procedure: Among the ensemble climate models, one climate model that is representing the median for most of the simulated hydrological components is selected for our current research. So, indeed, we did "run a number of climate models and then the results are averaged (in the case of impact models the input from a number of climate models are used and then results of the impact models are averaged)". This description will be added to the revised manuscript and we will correct the references to the cited studies as suggested by the reviewer.

However, we acknowledge that we did not propagate the impact of the different streamflow changes through the RVA analysis for being able to properly present the combinations of scenarios. To nevertheless address the raised concerns of uncertainty, we will additionally analyse and discuss the impacts of the min and max climate models —,which lead to the driest and wettest climate condition based on the simulated hydrologic parameters— from the full ensemble on groundwater recharge and the hydrological regime under different water consumptions and WUS scenarios.

When reading the paper Mahmoodi et al. (2020b) I also found that minimum and maximum elevation for the river basin shown in Figures 1 (Mahmoodi et al. (2020b) and the present study) are different - please explain.

Thank you for bringing this to our attention. This happened due to a wrong reference layer during figure preparation. We will correct this in the revised version.

In the whole text please use "streamflow" instead of "flow" when referring to river flow, otherwise readers could be confused as the paper is on surface water and groundwater. The results given for streamflow (IHA, VRA) are all for the outlet of the basin?

We will use streamflow and harmonize it throughout the manuscript and will explain that IHA and VRA analysis were carried out for the outlet of the basin.

**SPECIFIC COMMENTS:**

Minor comments regarding typos, rephrasing of sentences or adding of additional information on the model performance will all be addressed in the revised manuscript.

**Introduction:**

Page 1; Lines 23: "…alteration caused by natural or anthropogenic activities…" I am not sure what is "natural … activities"; rethink formulation.

We will revised it.

Page 2; Lines 40/41: "…precipitation... rainfall..."; please use either "precipitation" or "rainfall" in the whole text

We will use "precipitation" and harmonize it throughout the text.

Page 3; Line 68: "Further aggravation will put increasing pressure on the..."; "aggravation" of what (rethink formulation)?

We mean "further aggravation of climate change".

**Materials and methods:**

Page 3; Lines 80/81: "…increased over the last years at the outlet of the basin during the past 33 years..."; reformulate "…increased over the last years … during the past 33 years..."

Thank you. We will modify it accordingly.

Page 4; Lines 95/96: "Good performance for modeling daily streamflow values was achieved judged by a multi-metric approach including NSE (0.76 and 0.54)..."; according to Moriasi et al. (2007: Model evaluation guidelines for systematic quantification of accuracy in watershed

simulations. Transactions of the ASABE, 50(3): 885–900) an NSE of 0.54 is considered as "satisfactory".

Thank you, our statement is misleading. The NSE for the calibration period was 0.76 and for the validation period 0.54 at the daily time step. Since the time step in Moriasi et al. (2007) was monthly, our model performance can be rated as very good for the calibration period with NSE=0.81 and satisfactory for the validation period with NSE=0.6. We will add the performance rating separately for each calibration and validation periods in the revised version.

Page 4; Lines 114 and 116 (also line 459 "Statistics Center of Iran"): on page 114 it is "Statistics Cerner of Iran"; on page 116 it is "Statistical Center of Iran" - The homepage (https://www.amar.org.ir/english) gives the translation/name "Statistical Centre of Iran", please use the official translation/name given on the homepage.

Thank you for pointing this out. We will add the official name given on the homepage.

Page 4; Lines 123/124: "To meet the future domestic, agricultural and industrial water demand, increases in the number of wells and qanats are linearly extrapolated...", so you assume that all wells and qanats have the same (water) yield? Please justify this assumption.

The wells and qanats have specific water yields that have been derived from the available measurements provided by Iran Water & Power Resources Development Company (IWPCO) in 2001 and 2006 for the baseline scenario. There is no data on possible future extractions from wells and qanats and we therefore needed to define the most sensible extraction rates. By extrapolating the current rates linearly with population growth, we believe to have chosen a conservative scenario.

**Results:**

Page 7; Line 199: "...unsustainable subbasins (GWW<GWR)...", unsustainable is "(GWW>GWR)"; however, I suggest to change the wording (see "GENERAL COMMENTS") and this part needs to be rewritten anyway…

We will modify this in revised version. Please see our answer for the GERNAL COMMENTS section.

Page 7; Lines 207/208: "... Among these 56 unsustainable sub-basins, GWW/GWR ratio is higher than 5 in 42 sub-basins....", this means "only 20% of the water demand can be supplied sustainably"?; change the wording (see "GENERAL COMMENTS") - all the results presented need to be changed accordingly.

We will thoroughly revise the manuscript in this regard. Please see our answer for the GERNAL COMMENTS section.

Page 8; Lines 211/212: "...it drops from 385 (106 m3 yr-1) in model setup period to 172 (106 m3 yr-1)...", what is the unit of the first numbers (385; 172)?

The unit is "mill. $m^3$ $yr^{-1}$. The number "$10^6$" is changed to "mill." in the revised version of the manuscript.

Page 8; Lines 217: "3.1" is "Groundwater sustainability"; therefore I suggest to call "3.2" "Streamflow alteration",... give a subtitle that's clearly points to Streamflow/Surface water...

Thank you for the suggestion. We will modify it in the revised version.

**Discussion:**

Page 10; Line 282: "…substantial deficits in discharge during..."; is this "groundwater recharge" or "river discharge"?

It is river discharge.

Page 11; Line 285: "…predicted unsustainability of groundwater could..."; correct "…predicted unsustainability of groundwater use could..."?

Thank you. We will correct it.

Page 11; Line 287: "…could lead to a higher groundwater withdrawal in summer season..."; correct "…could lead to a higher demand on groundwater in summer season..."?

Thank you for the suggestion. We modified this to "could lead to higher demand on groundwater withdrawal in summer season when the surface water does not meet the rising demand.

**Conclusion:**
Needs to be adapted according to "GENERAL COMMENTS".

**Supplementary material:**
- in Figure S1 units (e.g. for "Group 1" and "Group 2" the for x-axis "year" and for the y-axis "m3 s-1") should be given

Thank you. We will add those to the revised version.

- Figure S2 needs a better description to understand what is shown (e.g. the title states "Distribution of annual values…" but shown are monthly values; or is it - the single dots - the 30 monthly values of the 30-years per period, e.g. for January?);

Each single dot representing the value of a specific month e.g. January, in a year. We will change the caption in the revised version.

also here units are missing. The results shown are scattered (not ordered) – is this because the results are given according to their temporal occurrence (i.e. the year)? If yes, why a different x-axis when compared to Fig. S1?

We used a different x-axis compared to Fig. S1 to provide a better overview of the changes in each IHA. The ordered values were already shown in Fig. S1. The shape of the violin plots illustrates the changes explicitly.

---

## Author Comment (AC2)

**Reply to Referee #2**

Dear Referee,
We appreciate the efforts you have invested on our manuscript. In the following, we listed your comments in black font together with our point-by-point response to each in blue color. In addition, we explained how each of the comments will be addressed in the revised manuscript.

Sincerely,
Nariman Mahmoodi, Jens Kiesel, Paul D. Wagner, Nicola Fohrer

**GENERAL COMMENTS:**

The topic covered fully corresponds to the aims and scope of the Hydrology and Earth System Sciences. Moreover, it covers aspects of hydrology and management. Though, the manuscript, in its present form, has a few weaknesses. Appropriate revisions to the following points should be undertaken to improve the readability and increase the interest for a general audience.

The title should better describe the study. Sustainable water use is not the subject of this study. I would rather say potential future climate impacts on water resources. At the end of the introduction provide a statement on the novelty of the study. At the moment it is not clear enough. Two sections of the manuscript need improvements. More specifically 2.2 Hydrological model and 2.3 Future climate change simulation sections. These two sections are parts of previously published work and they are confusing in relation to this research effort. Authors should better justify the selection of the G-RCM. I strongly recommend a flow chart of the methodology followed. Please provide as supplementary table the IHA Non-Parametric RVA Scorecard. Methodology limitations should be mentioned.

We appreciate the very insightful evaluation. The title is changed to "Spatially distributed impacts of climate change and ground water demand on the water resources in a Wadi system". The introduction is revised and novelty of the research is stressed. A novel aspect of our study is that while climate change impacts on water resources are well understood, the combined effects of climate change and population growth / water demand on water resources are rarely analyzed in a spatially distributed way. The methodology of the research is illustrated using a flow chart which will be added to the revised manuscript.

[Figure]

Figure1: Flow chart of the methodology employed.

The non-parametric IHA scorecard will be provided for the model setup and the future periods under different WUS scenarios (Table 1). These results will be added to the supplementary document with further explanation on the scorecard.

Table 1: Non-parametric IHA scorecard for the model setup and the future periods under the three WUS-scenarios.

| | Baseline period | Future period 1 | | | Future period 2 | | |
|---|---|---|---|---|---|---|---|
| | | No-WUS | Constant-WUS | Projected-WUs | No-WUS | Constant-WUS | Projected-WUs |
| *Normalization Factor* | 1 | 1 | 1 | 1 | 1 | 1 | 1 |
| *Mean annual flow ($m^3 \ s^{-1}$)* | 12.13 | 10.66 | 9.94 | 9.23 | 6.13 | 5.84 | 5.5 |
| *Non-Normalized Mean ($m^3 \ s^{-1}$) Flow* | 12.13 | 10.66 | 9.94 | 9.23 | 6.13 | 5.84 | 5.5 |
| *Annual C. V.* | 2.99 | 4.39 | 4.5 | 4.63 | 4.76 | 4.9 | 5.06 |
| *Flow predictability* | 0.28 | 0.32 | 0.34 | 0.34 | 0.43 | 0.44 | 0.44 |
| *Constancy/predictability* | 0.54 | 0.60 | 0.62 | 0.62 | 0.71 | 0.70 | 0.70 |
| *% of floods in 60d period* | 0.38 | 0.38 | 0.38 | 0.38 | 0.38 | 0.38 | 0.38 |
| *Flood-free season* | 10 | 48 | 48 | 48 | 124 | 124 | 124 |

Discussion section should be extended, as this is the main contribution to future researchers. Explain and discuss why the streamflow time series was generated for the period from 1993 to 2009 and 1979-2009. The presented results are influenced by this choice and it should be motivated. Moreover, at least a minimal discussion should be provided on expected differences (if any) when using different reference periods. Please contrast your own findings with those of previous literature about hydrological studies and climate change in the area (e.g. https://doi.org/10.1029/2008WR007615; https://doi.org/10.1186/s40068-019-0135-3; https://doi.org/10.1007/s10666-019-09665-x). Uncertainty of the model outputs should be also in discussion.

The time period of observed data does not match in terms of hydrologic and climatic input data in the study area, i.e., from 1993-2009 for streamflow and from 1979 to 2009 for climate components (precipitation, temperature, relative humidity, solar radiation, and wind speed). Therefore, the observed streamflow data from 1993 to 2009 is used to calibrate the model, then the calibrated model is run to simulate streamflow for the whole period (1979-2009) using the observed climate

data. Due to these data availability constraints, we cannot provide an analysis of the impact of different reference periods. We will however include a short discussion referring to other studies (such as e.g. Kiesel et al. (2019) where the impact of two different reference period lengths were evaluated). Thank you also for providing these additional references, which we will include in our discussion and relate to our results.

Reference:

Kiesel, J., Gericke, A., Rathjens, H., Wetzig, A., Kakouei, K., Jähnig, S.C. and Fohrer, N., 2019. Climate change impacts on ecologically relevant hydrological indicators in three catchments in three European ecoregions. Ecological engineering, 127, pp.404-416.

**Specific Comments**

All of the following minor comments regarding typos, rephrasing of sentences or adding of additional information on the model performance will be addressed in the revised manuscript.

**Comment 1**

Page 1, Line 11-13"A hydrologic model is used to assess streamflow and groundwater recharge of the Halilrood Basin on a daily time step under different scenarios over a model setup period (1979-2009) and for two future scenario periods (near future: 2030–2059 and far future: 2070-2099)." Please specify the number of the different scenarios.

We considered five different scenarios in this study. The information will be added to the abstract.

**Comment 2**

Page 3, Line 92-94"Based on representative climatic conditions, an eight-year period was used for model calibration (1995-2003) and a six-year period for validation (2004–2009)." Please specify which are the representative climatic conditions. Are these observations? If yes please provide the reference and the source of the data.

The model is calibrated using observed data provided by Iran Water & Power Resources Development Company (IWPCO). The calibration and validation periods are based on an equal distribution of dry years (total precipitation<200 mm), wet years (total precipitation>270 mm) and average year (200–270 mm annual precipitation) in the study area. Both periods are composed of almost 1/3 dry, wet, and normal years, respectively. This information will be added to the revised version.

**Comment 3**

Page 3, Line 88-89:"The Soil and Water Assessment Tool (SWAT, Arnold et al., 1998; Arnold et al., 2012) was used to simulate the streamflow of Halilrood River between 1993 and 2009 on a daily time step." Perhaps, authors mean that the outputs of the SWAT model were used in this study. This description is poor and confusing. Most of the work presented in this section, has been carried out in Mahmoodi et al. (2020a). Please consider to rewrite this section by clarifying the work you did for this research effort. Moreover, please specify if SWAT parametrization using the representative climatic conditions is sufficient when using the CSIRO-SMHI. In Mahmoodi et al. (2020b) it is mentioned that "Further details on model parameterization and performance are available in Mahmoodi et al. (2020)". I couldn't find this study on the internet. Mahmoodi N., Kiesel J., Wagner D. P. & Fohrer N. 2020 Integrating water use systems and soil and water conservation methods in a hydrological model of an Iranian Wadi system. J. Arid Land. 12 (4), 1–16

Indeed, we set up the model in a previous research effort and applied the model here to simulate different scenarios. We agree that it would be beneficial to add an overview of the model setup and calibration efforts to provide confidence in the model performance. We will modify this section and formulate more precisely. Running scenarios with a model calibrated for baseline conditions, is a common procedure in hydrologic impact assessments. The climatic conditions of the used and downscaled climate scenario CSIRO-SMHI are mostly within the range of conditions of the past (Figure 2). The driest future years which projected to occur more frequently are already included in the baseline years. Beyond that, SWAT is a process-based hydrological model that, unlike data-driven models, should be suitable for depicting the same processes under different (hotter and drier) boundary conditions (Zhu et al., 2016).

[Figure]

Figure 2: Annual precipitation ranges for the baseline (1993-2009) and the future periods

We are therefore confident that the model can sufficiently represent the future climate conditions. We are sorry to hear that the paper was not found and apologize for the inconvenience. Please find the paper following the link below (open access). https://doi.org/10.1007/s40333-020-0125-3

Reference:

Zhu, Q., Zhang, X., Ma, C., Gao, C. and Xu, Y.P., 2016. Investigating the uncertainty and transferability of parameters in SWAT model under climate change. Hydrological Sciences Journal, 61(5), pp.914-930.

**Comment 4**

Page 4, Line 110-112: "The calibrated and validated SWAT model was run with this climate model output to simulate groundwater recharge and streamflow for the model setup period (1979-2009) and two future periods (near future: 2030-2059 and far future: 2070-2099)". SWAT model was run in previous work (Mahmoodi et al., 2020b). Please, provide more information about the model outputs. Perhaps a table with descriptive statistical analysis of the observational streamflow data, together with the model outputs (e.g. streamflow for the three different periods) would help readers to follow the logic of the paper.

In the previous paper, we only simulated streamflow in the future. In this research, we combine a water demand scenario with a climate scenario. In addition, streamflow is simulated under pristine conditions. We will add a descriptive table to the revised manuscript, showing basic statistical analysis of streamflow at the basin outlet for baseline and scenario conditions (median, min-, and max- climate models).

Table 2: Statistical analysis of annual streamflow (m$^3$ s$^{-1}$) simulated for the model setup period (1979-2009) and two future periods (2030-2059 and 2070-2099) under Constant-WUS scenario for the median, max (wettest) and min (driest) climate models.

| | | | Median climate model | | Max climate model | | Min climate model | |
|---|---|---|---|---|---|---|---|---|
| | Observations | model setup period | Future period 1 | Future period 2 | Future period 1 | Future period 2 | Future period 1 | Future period 2 |
| | (1993-2009) | (1979-2009) | (2030-2059) | (2070-2099) | (2030-2059) | (2070-2099) | (2030-2059) | (2070-2099) |
| Mean | 7.66 | 13.31 | 9.93 | 5.84 | 11.62 | 15.11 | 2.85 | 2.13 |
| Max | 33.21 | 39.79 | 38.75 | 20.78 | 48.59 | 76.97 | 6.74 | 5.72 |
| Min | 0.43 | 1.56 | 0.53 | 0.85 | 0.34 | 0.21 | 0.77 | 0.3 |
| Median | 3.42 | 11.73 | 6.67 | 3.74 | 5.84 | 10.34 | 2.34 | 1.51 |
| STDEV | 8.20 | 10.27 | 9.77 | 5.34 | 13.58 | 17.20 | 1.72 | 1.48 |
| SKEW | 1.88 | 0.98 | 1.50 | 1.55 | 1.52 | 1.94 | 0.76 | 0.99 |

**Comment 5**

Page 4, Line 104-107 "For the present impact study, following the argumentation that errors level out and a projection can be better represented through averaging, i.e., taking the mean, median, or weighting (Tebaldi and Knutti 2007, Thober and Samaniego, 2014), from the RCP8.5 and distribution mapping-adjusted ensemble, one global-regional climate model was selected."

This sentence is a too long and it is confusing. Could you please explain further? Please also specify why you didn't use climate change projections from RCP 4.5.

We will modify the above sentences and clarify this statement in the revised version of the manuscript. We only used the RCP8.5 as one instructive and worst-case scenario. In addition, actual green-house gas emissions of the last decade have followed the RCP85 trajectory closer than the RCP45 scenario (Sanford et al., 2014). We will add these explanations to the revised version.

Reference:

Sanford, T., Frumhoff, P.C., Luers, A. and Gulledge, J., 2014. The climate policy narrative for a dangerously warming world. Nature Climate Change, 4(3), pp.164-166.

**Comment 6**

Page 4, Lines 107-108:"This selection was conducted according to the model democracy approach, which treats all climate models equally and the median model of the model ensemble is selected (IPCC, 2013)." What is the model democracy approach? In the IPCC 2013 there is no mention about it. Moreover, please fix the references according to the journal's requirements and in respect to the report's recommendations (see below). IPCC, 2013: Summary for Policymakers. In: Climate Change 2013: The Physical Science Basis. Contribution of Working Group I to the Fifth Assessment Report of the Intergovernmental Panel on Climate Change [Stocker, T.F., D. Qin, G. K. Plattner, M. Tignor, S.K. Allen, J. Boschung, A. Nauels, Y. Xia, V. Bex and P.M. Midgley (eds.)]. Cambridge University Press, Cambridge, United Kingdom and New York, NY, USA

The democracy approach is a synonym for the "ensemble of opportunity", a term which is used in the IPCC 2013 (chapter 11 and 12): It means that all models available define the ensemble and that each model has the same 'weight' or 'importance' (hence: democracy). From this full ensemble, the median model which is representing the median simulations for the hydrological components in the future periods (2030-2059 and 2070-2099) is selected for further investigation.

Thank you for providing the citation. It is modified accordingly in the reference list.

**Comment 7**

Page 4, Lines 108-110: "The G-RCM CSIRO-SMHI was chosen since it represented the median model of the major hydrological components (Mahmoodi et al., 2020b)." A model and its collection of runs is referred to as an ensemble. Please specify what do you mean as median model. If this ensemble selection influences the hydrological components in a major way, authors should add additional discussion in case of a different selection.

We believe that the focus on one climate model is instructive to discuss possible future impacts and the interaction of climate change impacts with the impacts of growing groundwater demand. We therefore determined a median model using the following procedure: Among the ensemble climate models, one climate model that is representing the median for most of the simulated hydrological components is selected for our current research. In our previous study, we addressed the inherent uncertainties in climate change assessment studies, using 11 climate models, 2 RCPs, and 2 bias correction methods in addition to the raw climate model data. Different climate models and projections will possibly lead to different results, as variability is shown in Figure 3 for each hydrological component under the ensemble of climate projections (Mahmoodi et al, 2020). We therefore can understand the concern that a different selection of a climate model may influence

the study's results and we will add an additional analysis of the full range of the climate model ensemble to the manuscript by showing the results for the min and max climate models.

[Figure]

Figure 3: Variability of hydrological components in the future under climate scenarios for the bias corrected—(LS: linear scaling in blue and DM: distribution mapping in red) and the raw data (Raw in yellow) compared to the historical period in gray (1979-2011). PCP: precipitation, ET: actual evapotranspiration, WYLD: water yield, SURQ: surface runoff, GWQ: groundwater flow, LATQ: lateral flow (Mahmoodi et al, 2020).

Reference:

Mahmoodi, N., Wagner, P.D., Kiesel, J. and Fohrer, N., 2020. Modeling the impact of climate change on streamflow and major hydrological components of an Iranian Wadi system. Journal of Water and Climate Change.

**Comment 8**

Page 4, Lines 117-119: "The population growth rate suggested by presidency of I.R.I, Plan and Budget Organization (2019) was applied on the 2017 population data to estimate the population

of the basin for the years 2045 and 2085, representative for the near and far future periods respectively (Table 1)."

If your analysis was made based on 2017 population data, why data relative to 2011 population are presented in tables (table 1 etc.)? If you didn't use these data please remove them.

The number of water use systems are reported until 2011. Therefore, we used the population growth rate between 2011 and 2017 to determine the number of WUS in 2017. We will correct this in the revision.

**Comment 9**

Page 4, Lines 123-124: "(i): To meet the future domestic, agricultural and industrial water demand, increases in the number of wells and qanats are linearly extrapolated with the estimated increases in the population of Halilrood Basin as follows:" Linear extrapolation should be discussed in the discussion and conclusion sections. Why authors decided to use this method?

Population growth is the main factor to predict the water consumption in Iran, as Keshavarz et al. (2006), reported a significant correlation between water consumption and population/size of households in Fars province with dry climate. In addition, the water consumption data reported for three provinces i.e., Azarbaijan, Khuzestan, Isfahan during the period 2001-2010 shows that the consumption rate is increasing linearly with population growth (Mombeni et al., 2013). Since the population and population growth rate are available for the study area, we assume that the current trend of population growth (current childbirth rate) will remain constant in the future. We believe that linking this increase (which probably overestimates future population) linearly with water demand, provides a rather conservative estimate of future water demand (we will discuss our results with reference to the population growth in Iran suggested by Presidency of I.R.I, Plan and Budget Organization). Therefore, using linear extrapolation is an applicable way to estimate the number of wells and qanats and water consumption in the future. We agree that other assumptions on population growth would lead to a different water demand.

References:

Keshavarzi, A.R., Sharifzadeh, M., Haghighi, A.K., Amin, S., Keshtkar, S. and Bamdad, A., 2006. Rural domestic water consumption behavior: A case study in Ramjerd area, Fars province, IR Iran. Water research, 40(6), pp.1173-1178.

Mombeni, H.A., Rezaei, S., Nadarajah, S. and Emami, M., 2013. Estimation of water demand in Iran based on SARIMA models. Environmental Modeling & Assessment, 18(5), pp.559-565.

**Comment 10**

Page 7, Lines 186-187:"The direction of change is shown by positive RVA, where the indicator becomes more stable within the RVA targets and negative RVA, where the indicator is moving towards an upper or lower alternative state." Please explain what do you mean "the indicator becomes more stable".

To evaluate the probable alteration through the RVA test, two targets, lower and upper, are considered. Stable situation is estimated for the indicator, whose variations stay within the targets. The phrase "becomes more stable" might be confusing; therefore, it will be changed to "where the indicator remains stable within the upper and lower bounds within the RVA targets and negative RVA, where the indicator is moving outside the upper or lower bounds to an alternative state."

**Comment 11**

Page 1, Line 42-42; Page 7, Line 203-204; Page 8, Line 225-226; Page 11, Line 287-288: "In Iran, the scarcity of rainfall, combined with climate change and population growth over the last decades, has resulted in higher groundwater extraction rates (Izady et al., 2015; Rafiei Emam et al., 2015; Mahmoudpour et al., 2016)." "As shown in Figure 2c and e, the GWW/GWR ratio is higher in the near and far future if the two stressors climate change and population growth are considered simultaneously (Projected-WUS)." "This might be due to the higher reduction in projected winter precipitation (Mahmoodi et al., 2020b). "This could lead to a higher groundwater withdrawal in summer season when the surface water does not meet the rising demand." Higher than what?

Thank you for pointing this out. We will modify those sentences in the revised version as follows:

"In Iran, groundwater extraction rates have increased over the last decades due to the scarcity of rainfall, combined with climate change and population growth (Izady et al., 2015; Rafiei Emam et al., 2015; Mahmoudpour et al., 2016)."

"As shown in Figure 2c and e, where the two stressors climate change and population growth are considered simultaneously (Projected-WUS), the GWW/GWR ratios in the near and far future are higher than the ratio during the model setup period."

"This might be due to the higher reduction in projected winter precipitation compared to the observations (Mahmoodi et al., 2020b)."

"This could lead to a higher groundwater demand in summer season as compared to the other seasons when the surface water does not meet the rising demand."

**Comment 12**

Page 5, Lines 135-136 "To disentangle the impacts of climate change and population growth and its combined effects on future aquifer condition and hydrologic regime, five scenarios were developed (Table 4)" I only see 4 scenarios in table 4. Please fix the table and also explain the symbol *.

Sorry for the confusion. "*" will be replaced by "✓" in the revised manuscript. The explanation for "✓" will be added below the table. "✓"addresses the scenario/s considered for each analysis. The table got accidentally truncated. We will make sure the size of the table matches so that all five scenarios fit on one page.

**Comment 13**

Page 9, Lines 236-237 "Although annual extreme flows mainly experience a lower degree of change in the near and far future (Figure 3 and 4),.." Should there be figure 4?

Thank you very much for spotting this. This should refer to Figure 3.

**Comment 14**

Table 8. Please write the relative years under each of the two future periods. Comment 15 Figure 3.

The relevant years will be added.

---

## Author Response (AR1)

**Comments from editor:**

Dear Authors,

As you are aware, the two reviewers have had a number of queries and recommendations to improve your manuscript. Both feel that the manuscript could be valuable if it is revised to address their concerns, but this will require major corrections. Please can you now incorporate your responses into a revised manuscript and resubmit it. We will then ask the reviewers to check that their concerns have been met.

**Response from authors:**

Dear Editor,

Thank you very much for the opportunity to revise our manuscript. All of the reviewer comments have been considered and incorporated in the revised version of the manuscript as described in our replies to the comments of the reviewers (AC1, AC2).

Kind regards

Nariman Mahmoodi

---

## Author Response (AR2)

Dear editor and referees,

We would like to express our gratitude towards the time and effort that you dedicated to providing valuable feedbacks to help in improving this journal paper. Below, we provide a point-by-point response (in yellow) to the reviewers' comments.

Sincerely,

Nariman

**Further amendments suggested by the reviewers are:**

- Figure 1 quality should be improved

Answer: We provided a figure with better resolution in the revised manuscript (Page21, L: 620).

**Abstract:**

Page 1; Line 8: replace "…withdrawal" by"… demand"

Answer: Thank you, changes applied.

**2. Materials and methods:**

Page 3; Lines 77/78: Change "Figure 1(b))... (Figure 1(b))... (Figure 1(c))" into" Figure 1b))... (Figure 1b))... (Figure 1c))" as in the remaining of the text.

Answer: These changes were made throughout the text.

Page 3; Lines 75-86: In the whole "2.1 Study area" description references are missing, e.g. sources of information/data on water demand/use, annual average precipitation/annual potential evaporation are from, GIS-data for springs, qanats, and wells…

Answer: We added the sources of data (Page3, L: 79, 80).

Page 3; Line 82: replace "Mahmoodi et al., (2020a)" by "Mahmoodi et al. (2020a)"

Answer: Thank you, changes applied.

Page 3; Line 92: replace "(Emami and Koch 2019;…" by "(Emami and Koch, 2019;…"

Answer: Thank you, changes applied.

Page 3; Lines 93/94: also here references for the data used for SWAT-setup (land-use, soils) are missing

Answer: The references were added (Page3, L:94).

Page 4; Lines 103/104: "…where the comparison showed a good agreement between simulated and observed PE (synoptic station, Figure 1d)." Reading this I was expecting a map/figure showing results for PE, instead the location of the synoptic station is shown. Maybe reformulate…"…where the comparison showed a good agreement between simulated and observed PE at the synoptic station shown in Figure 1d)."

Answer: Thank you for the suggestion. We revised it in the text.

Page 4; Lines 119/120: "RCP8.5 is selected since actual green-house gas emissions of the last decade have followed the RCP8.5 trajectory closer than any of the other RCPs (Sanford et al., 2014)." This statement was probably correct in 2014 – we are now in 2021 and to my understanding in recent year RCP8.5 has lost its importance in climate change impact studies because it is seen as unrealistic (see: https://www.nature.com/articles/d41586-020-00177-3). Think about reformulating…

Answer: Thank you for sharing this reference. Basically, we agree that the plausibility of some scenarios with high CO2 emissions, such as RCP8.5, can be discussed in light of recent developments in the energy sector. However as it is mentioned in IPCC (2021, Page 22-23) the projections from these scenarios can still be valuable. Accordingly, we reformulated this part in the text as "RCP8.5 is selected since actual green-house gas emissions of the last decade have followed the RCP8.5 trajectory closer than any of the other RCPs (Sanford et al., 2014). Although the projections driven from scenarios with high CO2 emissions (e.g. RCP8.5) have been criticized as mitigation measures are not accounted for (Hausfather and Peters, 2020), the projections from these scenarios can still be valuable (IPCC, 2021) as they not only agree with historical total cumulative CO2 emissions, but are also plausible for future projections given current and stated policies (Schwalm et al., 2020)".

*Reference*

*Hausfather, Z. and Peters, G.P.: Emissions–the 'business as usual'story is misleading. Nature, 577, 618-620, https://doi.org/10.1038/d41586-020-00177-3, 2020.*

*IPCC, 2021: Summary for Policymakers. In: Climate Change 2021: The Physical Science Basis. Contribution of Working Group I to the Sixth Assessment Report of the Intergovernmental Panel on Climate Change [Masson-Delmotte, V., P. Zhai, A. Pirani, S. L. Connors, C. Péan, S. Berger, N. Caud, Y. Chen, L. Goldfarb, M. I. Gomis, M. Huang, K. Leitzell, E. Lonnoy, J.B.R. Matthews, T. K. Maycock, T. Waterfield, O. Yelekçi, R. Yu and B. Zhou (eds.)]. Cambridge University Press. In Press.*

*Schwalm, C.R., Glendon, S. and Duffy, P.B.: RCP8. 5 tracks cumulative CO2 emissions. Proc. Natl. Acad. Sci., 117, 19656-19657, https://doi.org/10.1073/pnas.2007117117, 2020.*

Page 5; Lines 130/131: "Therefore, it can be assumed that the parameterized SWAT model can sufficiently represent the future climate conditions." Change to "Therefore, it can be assumed that the parameterized SWAT model is sufficiently applicable the future climate conditions."

Answer: Thank you. We modified it in the text.

Page 7; Lines 207/208 & 219: change "Vu et al., (2019) ... Richter et al., (1997)" to "Vu et al. (2019) ... Richter et al. (1997)"

Answer: Thank you, changes applied.

**3. Results:**

Here you should decide either to introduce the abbreviations GWD and GWR, and use these afterwards or not to introduce/ use these abbreviations. At the moment it is a mixture, e.g. "groundwater demand is higher than GWR." (page 8, line 230).

Answer: Thank you for pointing this out. We used the abbreviations for the sections 3.1.1 and 3.1.2.

Furthermore, in the text there is a mixture of tenses. While usually present is used, sometimes past tense is used, e.g. "...indicators have been significantly..." (page 9, line 260), "...21 IHA have indicated significant..." (page 12, line 359); change to "...indicators are significantly...", "...21 IHA indicate significant..." (check the whole text!).

Answer: Thank you for bring this to our attention. We used the present tense and harmonized it throughout the text.

Page 9; Line 278: "...none of minimum streamflows are expected..." change to "...none of minimum streamflow indicators is expected..."

Answer: Thank you, changes applied.

Page 10; Line 296: "...by around two month..." change to "...by around two months..."

Answer: Thank you, changes applied.

Page 10; Line 308: "...WUS in near and far future, are similar..." change to "...WUS in the near and far future are similar..."

Answer: Thank you, changes applied.

**4. Discussion:**

Page 12; Line 350: "Shahid et al., (2018)" change to "Shahid et al. (2018)"

Answer: Thank you, changes applied.

Page 12; Line 356: "…indicators are representing substantial…" change to …indicators show

Answer: Thank you, changes applied.